# End-to-end Learnable Clustering
# for Intent Learning in Recommendation

**Yue Liu**[*]
Ant Group
National University of Singapore
`yueliu19990731@163.com`

**Shihao Zhu**[*]
Ant Group
Hangzhou, China

**Jun Xia**
Westlake University
Hangzhou, China

**Yingwei Ma**
Alibaba Group
Hangzhou, China

**Jian Ma**
Ant Group
Hangzhou, China

**Xinwang Liu**[†]
National University of Defense Technology
Changsha, China

**Shengju Yu**
National University of Defense Technology
Changsha, China

**Kejun Zhang**
Zhejiang University
Hangzhou, China

**Wenliang Zhong**[†]
Ant Group
Hangzhou, China

## Abstract

Intent learning, which aims to learn users' intents for user understanding and item recommendation, has become a hot research spot in recent years. However, existing methods suffer from complex and cumbersome alternating optimization, limiting performance and scalability. To this end, we propose a novel intent learning method termed ELCRec, by unifying behavior representation learning into an End-to-end Learnable Clustering framework, for effective and efficient Recommendation. Concretely, we encode user behavior sequences and initialize the cluster centers (latent intents) as learnable neurons. Then, we design a novel learnable clustering module to separate different cluster centers, thus decoupling users' complex intents. Meanwhile, it guides the network to learn intents from behaviors by forcing behavior embeddings close to cluster centers. This allows simultaneous optimization of recommendation and clustering via mini-batch data. Moreover, we propose intent-assisted contrastive learning by using cluster centers as self-supervision signals, further enhancing mutual promotion. Both experimental results and theoretical analyses demonstrate the superiority of ELCRec from six perspectives. Compared to the runner-up, ELCRec improves NDCG@5 by 8.9% and reduces computational costs by 22.5% on the Beauty dataset. Furthermore, due to the scalability and universal applicability, we deploy this method on the industrial recommendation system with 130 million page views and achieve promising results. The codes are available on GitHub[3]. A collection (papers, codes, datasets) of deep group recommendation/intent learning methods is available on GitHub[4].

---

[*]Equal Contribution

[†]Corresponding Author

[3]https://github.com/yueliu1999/ELCRec

[4]https://github.com/yueliu1999/Awesome-Deep-Group-Recommendation

38th Conference on Neural Information Processing Systems (NeurIPS 2024).

# 1 Introduction

Sequential recommendation (SR), which aims to recommend relevant items to users by learning patterns from users' historical behavior sequences, is a vital and challenging task in the machine learning domain. In recent years, benefiting the strong representation learning ability of deep neural networks (DNNs), DNN-based sequential recommendation methods[105, 39, 94, 129, 50, 108, 52, 67] have achieved promising recommendation performance and attracted researchers' high level of attention.

More recently, intent learning has become a hot topic in both research and industrial field of recommendation. It aims to model users' intents by learning from users' historical behaviors. For example, a user interacts with shoes, bags, and rackets in history. Thus, the user's potential intent can be inferred as playing badminton. Then, the system may recommend the intent-relevant items to the user. Following this principle, various intent learning methods [44, 14, 45, 18, 49, 53, 5, 54] have been proposed to achieve better user understanding and item recommendation.

The optimization paradigm of recent representative intent learning methods can be summarized as a generalized Expectation Maximization (EM) framework. To be specific, at the E-step, clustering algorithms are adopted to learn the latent intents from users' behavior embeddings. In addition, in the M-step, self-supervised learning methods are utilized to embed behaviors. The optimizations of these two steps are performed alternately, achieving promising performance.

However, we highlight two issues in this complex and tedious alternating optimization. (1) At the E-step, we need to apply the clustering algorithm on the whole data, limiting the model's scalability, especially in large-scale industrial scenarios, e.g., apps with billion users. (2) In the EM framework, the optimization of behavior learning and the clustering algorithm are separated, leading to sub-optimal performance and increasing the implementation difficulty.

To this end, we propose a novel intent learning model named ELCRec via integrating representation learning into an End-to-end Learnable Clustering framework, for effective and efficient Recommendation. Specifically, the user's behavior process is first embedded into the latent space. Cluster centers, recognized as users' latent intents, are initialized as learnable neural network parameters. Then, a simple yet effective learnable clustering module is proposed to decouple users' complex intents into different simple intent units by separating the cluster centers. Meanwhile, it makes the behavior embeddings close to cluster centers to guide the models to learn more accurate intents from users' behaviors. This improves the model's scalability and alleviates issue (1) by optimizing the cluster distribution on mini-batch data. Furthermore, to further enhance the mutual promotion of representation learning and clustering, we present intent-assisted contrastive learning to integrate the cluster centers as self-supervision signals for representation learning. These settings unify behavior learning and clustering optimization in an end-to-end optimizing framework, improving recommendation performance and simplifying deployment. Therefore, issue (2) has also been solved. The contributions of this paper are summarized as follows.

- We innovatively promote the existing optimization framework of intent learning by unifying behavior representation learning and clustering optimization.

- A new intent learning model termed ELCRec is proposed with a simple yet effective learnable cluster module and intent-assisted contrastive learning.

- Comprehensive experiments and theoretical analyses show the advantages of ELCRec from six aspects, including superiority, effectiveness, efficiency, sensitivity, convergence, and visualization.

- We successfully deployed it on an industrial recommendation system with 130 million page views and achieved promising results, providing various practical insights.

# 2 Related Work

We provide a brief overview of the related work for this paper. It can be divided into three parts, including sequential recommendation, intent learning, and clustering algorithms. At first, Sequential Recommendation (SR) focuses on recommending relevant items to users based on their historical behavior sequences. In addition, intent learning has emerged as a promising and practical technique in recommendation systems. It aims to capture users' latent intents to achieve better user understanding

and item recommendation. Lastly, clustering algorithms play a crucial role in recommendation systems since they can identify patterns and similarities in the users or items. Due to the limitation of the pages, we introduce the detailed related methods in the Appendix 8.11.

# 3 Methodology

We present our proposed framework, ELCRec, in this section. Firstly, we provide the necessary notations and task definition. Secondly, we analyze and identify the limitations of existing intent learning. Finally, we propose our solutions to address these challenges. Before introducing our method, we first provide the intuitions and insights of designing ELCRec. Concretely, we first analyze the challenge of scaling the intent learning methods to large-scale industrial data. The existing intent learning methods always adopt the expectation and maximization framework, where E-step and M-step are conducted alternately and mutually promote each other. However, we find the EM framework is hard to scale to large-scale data since it faces two challenges. First, the clustering algorithm is performed on the full data, easily leading to the out-of-memory problem. Second, the EM paradigm limits performance since it separates the behavior learning process and the intent learning process. To solve these two problems, we aim to propose a new intent learning method for the recommendation task. For the first challenge, our initial idea is to design an online clustering method to update the clustering centers at each step. Specifically, we propose an end-to-end learnable clustering module (ELCM) to solve this problem by setting the clustering center as the learnable neural parameters and the pull-and-push cluster loss functions. In addition, for the second challenge, we aim to integrate the intent learning process into the behavior learning process and optimize them together. Benefitting from setting the cluster centers as the learnable neural parameters, we can utilize them to assist the behavior contrastive learning. Namely, we propose intent-assisted contrastive learning, which not only supports the learning process of online clustering but also unifies behavior learning and intent learning. Therefore, with the above two designs, we can solve the challenges of scaling the intent learning method to large-scale data.

## 3.1 Basic Notation

In a recommendation system, $\mathcal{U}$ denotes the user set, and $\mathcal{V}$ denotes the item set. For each user $u \in \mathcal{U}$, the historical behaviors are described by a sequence of interacted items $S^u = [s_1^u, s_2^u, ..., s_t^u, ..., s_{|S^u|}^u]$. $S^u$ is sorted by time. $|S^u|$ denotes the interacted items number of user $u$. $s_t^u$ denotes the item which is interacted with user $u$ at $t$ step. In practice, during sequence encoding, the historical behavior sequences are limited with a maximum length $T$ [34, 39, 18]. The sequences are truncated and remain the most recent $T$ interacted items if the length is greater than $T$. Besides, the shorter sequences are filled with "padding" items on the left until the length is $T$. Due to the limitation of the pages, we list the basic notations in Table 5 of the Appendix 8.2.

## 3.2 Task Definition

Given the user set $\mathcal{U}$ and the item set $\mathcal{V}$, the recommendation system aims to precisely model the user interactions and recommend items to users. Take user $u$ for an example, the sequence encoder firstly encodes the user's historical behaviors $S^u$ to the latent embedding $\mathbf{E}^u$. Then, based on the historical behavior embedding, the target of the recommendation task is to predict the next item that is most likely interacted with by user $u$ at $|S^u| + 1$ step.

## 3.3 Problem Analyses

Among the techniques in recommendation, intent learning has become an effective technique for understanding users. We summarize the optimization procedure of the intent learning as the Expectation Maximization (EM) framework. It contains two steps including E-step and M-step. These two steps are conducted alternately, mutually promoting each other. However, we find two issues of the existing optimization framework as follows.

(1) In the process of E-step, it needs to perform a clustering algorithm on the full data, easily leading to out-of-memory or long-running time problems. It restricts the scalability of the model on large-scale industrial data.

(2) The alternative optimization approach within the EM framework separates the learning process for behaviors and intents, leading to sub-optimal performance and increased implementation complexity. Also, it limits the training and inference of real-time data. That is, when users' behaviors and intents change over time, there is a long lag in the training and inference process.

Therefore, we aim to develop a new optimization framework for intent learning to solve issues (1) and issues (2). For issue (1), a new learnable online clustering method is the key solution. For the issue (2), we aim to break the alternative optimization in the EM framework.

## 3.4 Proposed Method

To this end, we present a new intent learning method termed ELCRec by unifying sequence representation learning into an End-to-end Learnable Clustering framework, for Recommendation. It contains three parts, including behavior encoding, end-to-end learnable cluster module (ELCM), and intent-assisted contrastive learning (ICL).

### 3.4.1 Behavior Encoding

In this process, we aim to encode the users' behavior sequences. Concretely, given the user set $\mathcal{U}$, the item set $\mathcal{V}$, and the users' historical behavior sequence set $\{S^u\}_{u=1}^{|\mathcal{U}|}$, the behavior encoder $\mathcal{F}$ embeds the behavior sequences of each user $u$ into the latent space as follows.

$$\mathbf{E}^u = \mathcal{F}(S^u), \tag{1}$$

where $\mathbf{E}^u \in \mathbb{R}^{|S^u| \times d'}$ denotes the behavior sequence embedding of user $u$, $d'$ is the dimension number of latent features, and $|S^u|$ denotes the length of behavior sequence of user $u$. Note that the behavior sequence lengths of different users are different. Therefore, all user behavior sequences are pre-processed to the sequences with the same length $T$ by padding or truncating. The encoder $\mathcal{F}$ is designed as a Transformer-based [100] architecture. Subsequently, to summarize the behaviors over different times of each user, the behavior sequence embedding is aggregated by the concatenate pooling function $\mathcal{P}$ as follows.

$$\mathbf{h}_u = \mathcal{P}(\mathbf{E}^u) = \text{concat}(\mathbf{e}_1^u || ... \mathbf{e}_i^u ... || \mathbf{e}_T^u), \tag{2}$$

where $\mathbf{e}_i^u \in \mathbb{R}^{1 \times d'}$ denotes the embedding of user behavior at $i$-th step and $\mathbf{h}_u \in \mathbb{R}^{1 \times Td'}$ denotes the aggregated behavior embedding of user $u$. We re-denote $Td'$ as $d$ for convenience. By encoding and aggregation, we obtain the behavior embeddings of all users $\mathbf{H} \in \mathbb{R}^{|\mathcal{U}| \times d}$.

### 3.4.2 End-to-end Learnable Cluster Module

After behavior encoding, we guide the model to learn the users' latent intents from the behavior embeddings. To this end, an end-to-end learnable cluster module (ELCM) is proposed to break the alternative optimization in the previously mentioned EM framework. This module can group the users' behaviors embeddings into various clusters, which represent the users' latent intents or interests. Concretely, at first, the cluster centers $\mathbf{C} \in \mathbb{R}^{k \times d}$ are initialized as the learnable neural parameters, i.e., the tensors with gradients. Then, we design a simple yet effective clustering loss to train the networks and cluster centers as formulated as follows.

$$\mathcal{L}_{\text{cluster}} = \underbrace{\frac{-1}{(k-1)k} \sum_{i=1}^{k} \sum_{j=1, j \neq i}^{k} \|\hat{\mathbf{c}}_i - \hat{\mathbf{c}}_j\|_2^2}_{\text{Intent Decoupling}} + \underbrace{\frac{1}{bk} \sum_{i=1}^{b} \sum_{j=1}^{k} \left\|\hat{\mathbf{h}}_i - \hat{\mathbf{c}}_j\right\|_2^2}_{\text{Intent-behavior Alignment}}, \tag{3}$$

where $\hat{\mathbf{h}}_i = \mathbf{h}_i / \|\mathbf{h}_i\|_2, \hat{\mathbf{c}}_i = \mathbf{c}_i / \|\mathbf{c}_i\|_2$. In Eq. (3), $k$ denotes the number of clusters (intents), and $b$ denotes the batch size. $\mathbf{h}_i \in \mathbb{R}^{1 \times d}$ denotes the $i$-th user's behavior embedding and $\mathbf{c}_j \in \mathbb{R}^{1 \times d}$ denotes the $j$-th cluster center. For better network convergence, we constrain the behavior embeddings and cluster center embeddings to distribute on a unit sphere. Concretely, we apply the $l$-2 normalization to both the user behavior embeddings $\mathbf{H}$, and the cluster centers $\mathbf{C}$ during calculating $\mathcal{L}_{\text{cluster}}$.

In the proposed clustering loss, the first term is designed to disentangle the complex users' intents into simple intent units. Technically, it pushes away different cluster centers, therefore reducing the

overlap between different clusters (intents). The time complexity and space complexity of this term are $\mathcal{O}(k^2 d)$ and $\mathcal{O}(kd)$, respectively. The number of users' intents is vastly less than the number of users, i.e., $k \ll |\mathcal{U}|$. Therefore, the first term will not bring significant time or space costs.

In addition, the second term of the proposed clustering loss aims to align the users' latent intents with the behaviors by pulling the behavior embeddings to the cluster centers. This design makes the in-class cluster distribution more compact and guides the network to condense similar behaviors into one intention. Also, on another aspect, it forces the model to learn users' intents from behavior embeddings. Note that the behavior embedding $\mathbf{h}_i$ is pulled to all center centers $\mathbf{c}_j, j = 1, ..., k$ rather than the nearest cluster center. The main reason is that the practical clustering algorithm is imperfect, and pulling to the nearest center easily leads to the confirmation bias problem [75]. To this end, the proposed clustering loss $\mathcal{L}_{\text{cluster}}$ aims to optimize the clustering distribution in an adversarial manner by pulling embeddings together to cluster centers while pushing different cluster centers away. Besides, it enables the optimization of this term via mini-batch samples, avoiding performance clustering algorithms on the whole data. The time complexity and space complexity of the second term are $\mathcal{O}(bkd)$ and $\mathcal{O}(bk + bd + kd)$, respectively. Since the batch size is essentially less than the number of users, namely, $b \ll |\mathcal{U}|$, the second term of clustering loss $\mathcal{L}_{\text{cluster}}$ alleviates the considerable time or space costs.

In the existing EM optimization framework, the clustering algorithm needs to be applied on the entire users' behavior embeddings $\mathbf{H} \in \mathbb{R}^{|\mathcal{U}| \times d}$. Take the classical $k$-Means clustering as an example, at each E-step, it leads to $\mathcal{O}(t|\mathcal{U}|kd)$ time complexity and $\mathcal{O}(|\mathcal{U}|k + |\mathcal{U}|d + kd)$ space complexity, where $t$ denote the iteration steps of $k$-Means clustering algorithm. We find that, at each step, the time and space complexity is linear to the number of users, thus leading to out-of-memory or running time problems (issue (1)), especially on large-scale industrial data with millions or billions of users.

Fortunately, our proposed end-to-end learnable cluster module can solve this issue (1). By summarising previous analyses, we draw that the overall time and space complexity of calculating the clustering loss $\mathcal{L}_{\text{cluster}}$ are $\mathcal{O}(bkd + k^2 d + bd)$ and $\mathcal{O}(bk + bd + kd)$, respectively. They are both linear to the batch size $b$ at each step, enabling the model's scalability. Besides, the proposed module is plug-and-play and easily deployed in real-time large-scale industrial systems. We provide detailed evidence and practical insights in Section 5. The proposed ELCM can not only improve the recommendation performance (See Section 4.1 & 4.2) but also promote efficiency (See Section 4.3).

### 3.4.3 Intent-assisted Contrastive Learning

Next, we aim to enhance further the mutual promotion of behavior learning and clustering. To this end, Intent-assisted contrastive learning (ICL) is proposed by adopting cluster centers as self-supervision signals for behavior learning. Firstly, we conduct contrastive learning among the behavior sequences. The new views of the behavior sequences are constructed via sequential augmentations, including mask, crop, and reorder. The two views of behavior sequence of user $u$ are denoted as $(S^u)^{v1}$ and $(S^u)^{v2}$. According to Section 3.4.1, the behaviors are encoded to the behavior embeddings $\mathbf{h}_u^{v1}, \mathbf{h}_u^{v2} \in \mathbb{R}^{1 \times d}$. Then, the sequence contrastive loss of user $u$ is formulated as follows.

$$\mathcal{L}_{\text{seq\_cl}}^u = -\left( \log \frac{e^{\text{sim}(\mathbf{h}_u^{v1}, \mathbf{h}_u^{v2})}}{\sum_{\text{neg}} e^{\text{sim}(\mathbf{h}_u^{v1}, \mathbf{h}_{\text{neg}})}} + \log \frac{e^{\text{sim}(\mathbf{h}_u^{v1}, \mathbf{h}_u^{v2})}}{\sum_{\text{neg}} e^{\text{sim}(\mathbf{h}_u^{v2}, \mathbf{h}_{\text{neg}})}} \right), \tag{4}$$

where "sim" denotes the dot-product similarity, "neg" denotes the negative samples. Here, the same sequence with different augmentations is recognized as the positive sample pairs, and the other sample pairs are recognized as the negative sample pairs. By minimizing $\mathcal{L}_{\text{seq\_cl}} = \sum_u \mathcal{L}_{\text{seq\_cl}}^u$, the similar behaviors are pulled together, and the others are pushed away from each other, therefore enhancing the representation capability of users' behaviors. The learned cluster centers $\mathbf{C} \in \mathbb{R}^{k \times d}$ are adopted as the self-supervision signals. The index of the assigned cluster of $\mathbf{h}_u^{v1}$ is queried as follows.

$$idx = \arg\min_i (\|\mathbf{c}_i - \mathbf{h}_u^{v1}\|_2^2), \tag{5}$$

where $\mathbf{c}_i \in \mathbb{R}^{1 \times d}$ denotes the $i$-th cluster (intent) center embedding. Then, the intent information is fused to the user behavior during the sequence contrastive learning. Here, we consider two optional fusion strategies, including the concatenate fusion $\mathbf{h}_u^{v1} = \text{concat}(\mathbf{h}_u^{v1} \| \mathbf{c}_{idx})$ and the shift fusion $\mathbf{h}_u^{v1} = \mathbf{h}_u^{v1} + \mathbf{c}_{idx}$. A similar operation is applied to the second view of the behavior embedding $\mathbf{h}_u^{v2}$. After fusing the intent information to user behaviors, the networks are trained by minimizing $\mathcal{L}_{\text{seq\_cl}}$.

In addition, to further collaborate intent learning and sequential representation learning, we conduct contrastive learning between the user's behaviors and the learnable intent centers. The intent contrastive loss is formulated as follows.

$$\mathcal{L}_{\text{intent\_cl}}^{u} = -\left( \log \frac{\min_i e^{\text{sim}(\mathbf{h}_u^{v1}, \mathbf{c}_i)}}{\sum_{\text{neg}} e^{\text{sim}(\mathbf{h}_u^{v1}, \mathbf{c}_{\text{neg}})}} + \log \frac{\min_i e^{\text{sim}(\mathbf{h}_u^{v2}, \mathbf{c}_i)}}{\sum_{\text{neg}} e^{\text{sim}(\mathbf{h}_u^{v2}, \mathbf{c}_{\text{neg}})}} \right), \tag{6}$$

where $\mathbf{h}_u^{v1}, \mathbf{h}_u^{v2}$ are two-view behavior embedding of the user $u$. Besides, "neg" denotes the negative behavior-intent pairs among all pairs. Here, we regard the behavior embedding and the corresponding nearest intent center as the positive pair and others as negative pairs. By minimizing the intent contrastive loss $\mathcal{L}_{\text{intent\_cl}} = \sum_u \mathcal{L}_{\text{intent\_cl}}^{u}$, behaviors with the same intents are pulled together, but behaviors with different intents are pushed away. The objective of ICL is formulated as follows.

$$\mathcal{L}_{\text{icl}} = \mathcal{L}_{\text{seq\_cl}} + \mathcal{L}_{\text{intent\_cl}}. \tag{7}$$

The effectiveness of ICL is verified in Section 4.2. With the proposed ELCM and ICL, we develop a new end-to-end optimization framework for intent learning, improving performance and convenience. By these designs, the issue (2) is also solved.

### 3.4.4 Overall Objective

The neural networks and learnable clusters are trained with multiple tasks, including intent learning, intent-assisted contrastive learning, and next-item prediction. The intent learning task aims to capture the users' underlying intents. Besides, intent-assisted contrastive learning aims to collaborate with intent learning and behavior learning. In addition, the next-item prediction task is a widely used task for recommendation systems. The overall objective of ELCRec is formulated as follows.

$$\mathcal{L}_{\text{overall}} = \mathcal{L}_{\text{next\_item}} + 0.1 \times \mathcal{L}_{\text{icl}} + \alpha \times \mathcal{L}_{\text{cluster}}, \tag{8}$$

where $\mathcal{L}_{\text{next\_item}}$, $\mathcal{L}_{\text{icl}}$, and $\mathcal{L}_{\text{cluster}}$ denotes the next item prediction loss, intent-assisted contrastive learning loss, and clustering loss, respectively. $\alpha$ is a trade-off hyper-parameter. We present the overall algorithm process of the proposed ELCRec method in Algorithm 1 in Appendix.

We detail and summarize the devised loss in equation (8). We train our proposed ELCRec method with multiple tasks, including the next-item prediction task, intent-assisted contrastive learning, and intent learning (learnable clustering) task. Accordingly, Equation (8), which denotes the overall loss function of ELCRec, contains three parts: next-item prediction loss $\mathcal{L}_{\text{next\_item}}$, the intent-assisted contrastive learning loss $\mathcal{L}_{\text{icl}}$, and the intent learning loss $\mathcal{L}_{\text{cluster}}$. Concretely, the next-item prediction loss is a commonly used loss function for the sequential recommendation. It aims to predict the next item in the interaction sequence based on the previous sequence. In addition, the intent learning loss aims to optimize the cluster center embeddings by pulling the samples to the corresponding cluster centers and pushing away different cluster centers. Moreover, the intent-assisted contrastive learning loss aims to conduct self-supervised learning to unify the behavior representation learning and intent representation learning. Overall, equation (8) trains the network through three tasks by a linear combination of three loss functions.

## 4 Experiment

This section aims to comprehensively evaluate ELCRec by answering research questions (RQs).

  (i) Superiority: does it outperform the state-of-the-art sequential recommendation methods?

 (ii) Effectiveness: are the ELCM and ICL modules effective?

(iii) Efficiency: how about the time and memory efficiency of the proposed ELCRec?

(iv) Sensitivity: what is the performance of the proposed method with different hyper-parameters?

 (v) Convergence: have the loss function and recommendation performance converged?

(vi) Visualization: Can the visualized learned embeddings reflect the promising results?

We answer RQ(i), (ii), (iii) in Section 4.1, 4.2, 4.3, respectively. Due to the limited pages, RQ(iv), (v), (vi) are answered in the Appendix 8.6, 8.7, and 8.8 respectively.

Table 1: Recommendation performance on benchmarks. **Bold values** and underlined values denote the best and runner-up results. * indicates that, in the $t$-test, the best method significantly outperforms the runner-up with $p < 0.05$. "-" indicates models do not converge.

| Dataset | Metric | BPR-MF [88] | GRU4Rec [34] | Caser [96] | SASRec [39] | BERT4Rec [94] | DSSRec [69] | S3-Rec [129] | CL4SRec [108] | DCRec [116] | MAERec [118] | IOCRec [49] | ICLRec [18] | ELCRec Ours | Impro. | p-value |
|---|---|---|---|---|---|---|---|---|---|---|---|---|---|---|---|---|
| **Sports** | HR@5 | 0.0141 | 0.0162 | 0.0154 | 0.0206 | 0.0217 | 0.0214 | 0.0121 | 0.0217 | 0.0172 | 0.0225 | 0.0246 | 0.0263 | **0.0286** | 8.75%↑ | 2.34e-6* |
| | HR@20 | 0.0323 | 0.0421 | 0.0399 | 0.0497 | 0.0604 | 0.0495 | 0.0344 | 0.0540 | 0.0357 | 0.0488 | 0.0641 | 0.0630 | **0.0648** | 1.09%↑ | 2.29e-4* |
| | NDCG@5 | 0.0091 | 0.0103 | 0.0114 | 0.0135 | 0.0143 | 0.0142 | 0.0084 | 0.0137 | 0.0118 | 0.0152 | 0.0162 | 0.0173 | **0.0185** | 6.94%↑ | 3.54e-5* |
| | NDCG@20 | 0.0142 | 0.0186 | 0.178 | 0.0216 | 0.0251 | 0.0220 | 0.0146 | 0.0227 | 0.0170 | 0.0225 | 0.0280 | 0.0276 | **0.0286** | 2.14%↑ | 7.87e-3* |
| **Beauty** | HR@5 | 0.0212 | 0.0111 | 0.0251 | 0.0374 | 0.0360 | 0.0410 | 0.0189 | 0.0423 | 0.0368 | 0.0414 | 0.0408 | 0.0495 | **0.0529** | 6.87% ↑ | 3.18e-6* |
| | HR@20 | 0.0589 | 0.0478 | 0.0643 | 0.0901 | 0.0984 | 0.0914 | 0.0487 | 0.0994 | 0.0674 | 0.0854 | 0.0916 | 0.1072 | **0.1079** | 0.65%↑ | 3.30e-3* |
| | NDCG@5 | 0.0130 | 0.0058 | 0.0145 | 0.0241 | 0.0216 | 0.0261 | 0.0115 | 0.0281 | 0.0269 | 0.0283 | 0.0245 | 0.0326 | **0.0355** | 8.90%↑ | 4.48e-6* |
| | NDCG@20 | 0.0236 | 0.0104 | 0.0298 | 0.0387 | 0.0391 | 0.0403 | 0.0198 | 0.0441 | 0.0357 | 0.0407 | 0.0407 | 0.0491 | **0.0509** | 3.67%↑ | 9.08e-6* |
| **Toys** | HR@5 | 0.0120 | 0.0097 | 0.0166 | 0.0463 | 0.0274 | 0.0502 | 0.0143 | 0.0526 | 0.0399 | 0.0477 | 0.0311 | **0.0586** | 0.0585 | 0.17%↓ | 1.22e-1 |
| | HR@20 | 0.0312 | 0.0301 | 0.0420 | 0.0941 | 0.0688 | 0.0975 | 0.0235 | 0.1038 | 0.0679 | 0.0904 | 0.0781 | 0.1130 | **0.1138** | 0.71%↑ | 4.20e-3* |
| | NDCG@5 | 0.0082 | 0.0059 | 0.0107 | 0.0306 | 0.0174 | 0.0337 | 0.0123 | 0.0362 | 0.0296 | 0.0336 | 0.0197 | 0.0397 | **0.0403** | 1.51%↑ | 2.87e-4* |
| | NDCG@20 | 0.0136 | 0.0116 | 0.0179 | 0.0441 | 0.0291 | 0.0471 | 0.0162 | 0.0506 | 0.0374 | 0.0458 | 0.0330 | 0.0550 | **0.0560** | 1.82%↑ | 3.72e-5* |
| **Yelp** | HR@5 | 0.0127 | 0.0152 | 0.0142 | 0.0160 | 0.0196 | 0.0171 | 0.0101 | 0.0229 | | 0.0166 | 0.0222 | 0.0233 | **0.0236** | 1.29% ↑ | 7.81e-3* |
| | HR@20 | 0.0346 | 0.0371 | 0.0406 | 0.0443 | 0.0564 | 0.0464 | 0.0314 | 0.0630 | - | 0.0460 | 0.0640 | 0.0645 | **0.0653** | 1.24%↑ | 3.73e-4* |
| | NDCG@5 | 0.0082 | 0.0091 | 0.0080 | 0.0101 | 0.0121 | 0.0112 | 0.0068 | 0.0144 | | 0.0105 | 0.0137 | 0.0146 | **0.0150** | 2.74%↑ | 1.23e-2* |
| | NDCG@20 | 0.0143 | 0.0145 | 0.0156 | 0.0179 | 0.0223 | 0.0193 | 0.0127 | 0.0256 | | 0.0186 | 0.0263 | 0.0261 | **0.0266** | 1.14%↑ | 6.82e-3* |

## 4.1 Superiority

In this section, we aim to answer the research question (i) and demonstrate the superiority of ELCRec. To be specific, we compare ELCRec with nine state-of-the-art recommendation baselines [88, 34, 96, 39, 69, 94, 129, 108, 18]. Experimental results are the mean values of three runs. As shown in Table 1, the **bold values** and underlined values denote the best and runner-up results, respectively. From these results, we have four conclusions as follows. (a) The non-sequential model BPR-MF [88] has not achieved promising performance since the shallow method lacks the representation learning capability of users' historical behaviors. (b) The conventional sequential methods [34, 96, 39] improve the recommendation via different DNNs such as CNN [42], RNN [121], and Transformer [100]. But they perform worse since limiting self-supervision. (c) The recent methods [94, 129, 108] enhance the self-supervised capability of models via the self-supervised learning techniques. However, they neglect the underlying users' intent, thus leading to sub-optimal performance. (d) More recently, the intent learning methods [44, 14, 45, 18, 49, 53, 5] have been proposed to mine users' underlying intent to assist recommendation. Motivated by their success, we propose a new intent learning method termed ELCRec. Befitting from the strong intent learning capability of ELCRec, it surpasses all other intent learning methods.

The balance is set to 1 in equation (7). We can add one balance hyperparameter to control the balance between sequence contrastive learning loss and intent contrastive learning loss to achieve better performance. However, in equation (8), we find there are many balances that need to be controlled, such as the balance of intent-assist contrastive learning loss and the balance of intent learning loss, easily leading to the high cost of hyperparameter tuning. To lower the load of tune hyperparameters, we fix the balance between sequence contrastive learning loss and intent contrastive learning loss as 1 and the balance between next item prediction loss and intent-assisted contrastive learning loss as 0.1. This setting has already been able to achieve promising performance. For other complex scenarios, we can set more balance hyperparameters for better performance in the future.

We did have one inconsistent finding on the toy dataset compared with other datasets. Concretely, ELCRec (B+ELCM+ICL) cannot beat B+ELCM, indicating that ICL may be ineffective on the B+ELCM variant on this dataset. However, we also find that B+ICL can beat B, indicating that ICL works for the baseline model. This phenomenon is interesting. We have the following explanations as follows. The ICL is conducted on both the behavior representations and the intent representations. Therefore, it can be influenced by both these two optimization processes. Namely, both the quality of behavior embeddings and the quality of the intent embeddings are crucial for the quality of ICL. Thus, it may not be very robust in all cases. For B+ICL, adding ICL to the baseline can improve the behavior-learning process. However, we find that B+ELCM has already achieved a very promising performance compared with other variants, indicating the quality of intent representations is excellent. Then we add ICL to B+ELCM, the ICL may downgrade the quality of intent representations. To solve this issue, we will conduct more careful training and optimize the training procedure to achieve better performance.

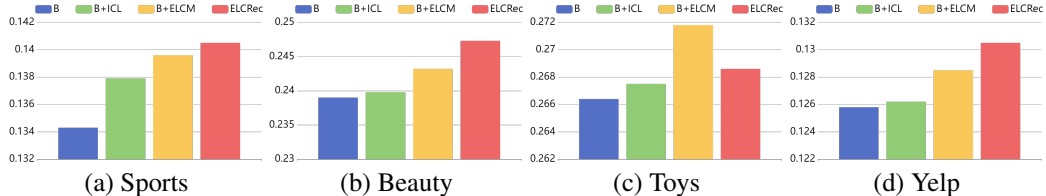

| (a) Sports | (b) Beauty | (c) Toys | (d) Yelp |

Figure 1: Ablation studies of the proposed end-to-end learnable cluster module (ELCM) and the intent-assisted contrastive learning (ICL). The results are the sum of four metrics, including HR@5, HR@20, NDCG@5, and NDCG@20.

To further verify the superiority of ELCRec, we conduct the $t$-test between the best and runner-up methods. As shown in Table 1, the most $p$-value is less than 0.05 except HR@5 on the Toys dataset. It indicates that ELCRec significantly outperforms runner-up methods. Overall, the extensive experiments demonstrate the superiority of ELCRec. In addition, we also conduct comparison experiments on recommendation datasets of other domains, including movie recommendation and news recommendation, as shown in the Appendix 8.4.1 and 8.4.2. These experimental results demonstrate a broader applicability of our proposed ELCRec.

## 4.2 Effectiveness

This section is dedicated to answering the research question (ii) and evaluating the effectiveness of the End-to-end Learnable Cluster Module (ELCM) and Intent-assisted Contrastive Learning (ICL). To achieve this, we conducted meticulous ablation studies on four benchmarks. Figure 1 illustrates the experimental results. In each sub-figure, "B", "B+ICL," "B+ELCM," and "ELCRec" correspond to the backbone, backbone with ICL, backbone with ELCM, and backbone with both ICL and ELCM, respectively. Through the ablation studies, we draw three key conclusions. (a) "B+ICL" outperforms the backbone "B" on all four benchmarks. It indicates that the proposed ICL effectively improves behavior learning. (b) "B+ELCM" surpasses the backbone "B" significantly on all benchmarks. This phenomenon demonstrates that our proposed end-to-end learnable cluster module helps the model better capture the users' underlying intents, thus improving recommendation performance. (c) ELCRec achieves the best performance on three out of four datasets. It shows the effectiveness of the combination of these two modules. On the Toys dataset, ELCRec can outperform the "B" and "B+ICL" but perform worse than "B+ELCM". This phenomenon indicates it is worth researching the better collaboration of these two modules in the future. To summarize, these extensive ablation studies verify the effectiveness of the proposed intent-assisted contrastive learning and end-to-end learnable cluster module in ELCRec.

## 4.3 Efficiency

We test the efficiency of ELCRec on four benchmarks and answer the research question (iii). Concretely, the efficiency contains two perspectives, including running time costs (in seconds) and GPU memory costs (in MB). Note that we use the same epoch number of our method and the baseline when we test the running time. Besides, we calculate the average GPU memory cost during the training process. We have two observations as follows. (a) ELCRec can speed up ICLRec on three out of four datasets (See Table 2). Overall, on four datasets, the running time is decreased by 7.18% on average. The reason is that our proposed end-to-end optimization of intent learning breaks the alternative optimization of the EM framework, saving computation costs. (b) The results demonstrate that the GPU memory costs of our ELCRec are lower than that of ICLRec on four datasets (See Table 2). On average, the GPU memory costs are decreased by 9.58%. It is because we enable the model to conduct intent learning via the mini-batch users' behaviors. Therefore, in summary, we demonstrate the efficiency of ELCRec from both time and memory aspects. Please note that, due to the relatively small size of the open benchmarks, the efficiency improvements are not particularly significant. However, on large-scale data, our method can achieve more substantial improvements.

We observe that in most cases, our proposed method can save time and memory costs, e.g., saving 7.18% time and 9.48% memory on average. For the time cost of our method on the Sports dataset, we regard it as a corner case. By careful analyses, we provide the explanation as follows. We suspect the raised time costs are caused by the wrong direction of the optimization. Setting the cluster

embeddings as the learnable neural parameters and optimizing them during training may be a harder task for the model compared to conducting the offline clustering algorithm on the learned embeddings directly. We analyze the performance and loss curve of our method on the Sports dataset and find that the decline of loss slowdowns and the performance seem to drop a little at almost the end of the training. We think this wrong optimization leads to the comparable time cost of our method compared with the baseline. But for other datasets, their optimization processes are great, therefore saving time and memory costs essentially. In the future, we can avoid this wrong optimization direction through some strategies, such as early-stopping and penalty terms.

Table 2: Running time and memory costs. **Bold values** denote better results.

| Cost | Dataset | Sports | Beauty | Toys | Yelp | Average |
|---|---|---|---|---|---|---|
| | ICLRec | 5282 | 3770 | 4374 | 4412 | 4460 |
| **Time** | ELCRec | 5360 | 2922 | 4124 | 4151 | 4139 |
| | Improvement | 1.48% ↑ | **22.49%** ↓ | **5.72%** ↓ | **5.92%** ↓ | **7.18%** ↓ |
| | ICLRec | 1944 | 1798 | 2887 | 3671 | 2575 |
| **Memory** | ELCRec | 1781 | 1594 | 2555 | 3383 | 2328 |
| | Improvement | **8.38%** ↓ | **11.35%** ↓ | **11.50%** ↓ | **7.85%** ↓ | **9.58%** ↓ |

# 5 Application

Our proposed ELCRec is versatility and plug-and-play. Benefiting its advantages, we aim to apply it to real-time large-scale industrial recommendation systems with millions of users. First, we introduce the background and settings of the application. Then, we conduct extensive A/B testing and analyze the experimental results. Besides, due to the page limitation, we provide deployment details and practical insights in Appendix 8.13 and 8.10, respectively.

## 5.1 Application Background

The applied scenario is the live streaming recommendation on the front page of the Alipay app. The user view (UV) and page view (PV) of this application are about 50 million and 130 million, respectively. Note that most users are new to this application, therefore leading to the sparsity of users' behaviors. To solve this cold-start problem in the recommendation system, we adopt our proposed method to group users and recommend items based on the groups. Concretely, due to the sparsity of users' behaviors, we first replace the users' behavior with the users' activities features in this application and model them via the multi-gate mixture-of-expert (MMOE) model [68]. Th,en we aim to group the users into various groups. For the existing intent learning methods, they easily lead to long-running time or out-of-memory problems. To solve this problem, we adopt the end-to-end learnable cluster module to group the users into various groups effectively and efficiently. Through this module, the high-activity users and new users are grouped into different clusters, alleviating the cold-start issue and assisting in better recommendations. Besides, during the learning process of the cluster embeddings, the low-activity users can transfer to high-activity users, improving the overall users' activities in the application. Eventually, the networks are trained with multiple tasks. In the next section, we conduct experiments to demonstrate the effectiveness of our proposed method on real-time large-scale industrial data.

## 5.2 A/B Testing on Real-time Large-scale Data

We conduct A/B testing on the real-time large-scale industrial recommendation system. The experimental results are listed in Table 3. We evaluate the models with two metric systems, including live streaming metrics and merchandise metrics. livestreaming metrics contain Page View Click Through Rate (PVCTR) and Video View (VV). Merchandise metrics contain PVCTR and User View Click Through Rate (UVCTR). The results indicate that our method can improve the recommendation performance of the baseline by about 2%. Besides, the improvements are significant with $p < 0.05$ in three out of four metrics.

Table 3: A/B testing on real-time large-scale industrial recommendation. **Bold values** denotes the significant improvements with $p < 0.05$. The symbol "-" denotes business secret.

| Method | Livestreaming Metrics | | Merchandise Metrics | |
|---|---|---|---|---|
| | PVCTR | VV | PVCTR | UVCTR |
| Baseline | - | - | - | - |
| Impro. | **2.45%** ↑ | **2.28%** ↑ | **2.41%** ↑ | 1.62% ↑ |

In addition, to further explore why our method can work well in real-time large-scale recommendation systems, we further analyze the recommendation performance of different user groups. The results are shown in Table 4. Based on the users' activity, we classify them into five groups, including Pure New users (PN), New users (N), Low-Activity users (LA), Medium-Activity users (MA), and High-Activity users (HA). Compared with the general recommendation algorithms that are unfriendly to new users, the experimental results show that our module not only improves the recommendation performance of high-activity users but also improves the recommendation performance of new users. Therefore, it can alleviate the cold-start problem and construct a more friendly user ecology.

For the utilization of group embeddings, there are many ways. For the conventional user recommendation or the group recommendation, we utilize the historical group embeddings and conduct continued training for the recommendation model. For other downstream tasks in other domains, we can provide the restore group embeddings for them. Therefore, for the recommendation model, the group embeddings are restored in the model parameters and updated daily. Besides, for other indirect downstream tasks, the group embeddings will be stored in the database.

Table 4: Results on different user groups. **Bold values** denotes improvements with $p < 0.05$.

| Metric | PN | N | LA | MA | HA |
|---|---|---|---|---|---|
| PVCTR | **6.96%** ↑ | **1.67%** ↑ | **1.98%** ↑ | **0.35%** ↑ | **19.02%** ↑ |
| VV | **6.81%** ↑ | **1.50%** ↑ | **1.50%** ↑ | **0.04%** ↑ | **16.90%** ↑ |

## 6 Conclusion

In this paper, we explore intent learning in recommendation systems. To be specific, we summarize and analyze two drawbacks of the existing EM optimization framework of intent learning. The complex and cumbersome alternating optimization limits the scalability and performance of existing methods. To this end, we propose a novel intent learning method termed ELCRec with an end-to-end learnable cluster module and intent-assisted contrastive learning. Extensive experiments on four benchmarks demonstrate ELCRec's six abilities. In addition, benefiting from the versatility of ELCRec, we successfully apply it to the real-time large-scale industrial scenario and also achieve promising performance. Due to the limited pages, We discuss the limitations and future work of this paper in Appendix 8.14, such as pre-defined cluster number, limited recommendation domains, and uncontrollable update rate of cluster centers.

## 7 Acknowledgment

We thank all anonymous reviewers for their constructive and helpful reviews. This work was supported by the National Natural Science Foundation of China (No. 62325604 and 62276271). Besides, this work was also supported by the National Key R&D Program of China (Project 2022ZD0115100), the National Natural Science Foundation of China (Project U21A20427), the Research Center for Industries of the Future (Project WU2022C043), and the Competitive Research Fund (Project WU2022A009) from the Westlake Center for Synthetic Biology and Integrated Bioengineering.

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

# 8 Appendix

## 8.1 Experimental Setup

### 8.1.1 Experimental Environment

Experimental results on the public benchmarks are obtained from the desktop computer with one NVIDIA GeForce RTX 4090 GPU, six 13th Gen Intel(R) Core(TM) i9-13900F CPUs, and the PyTorch platform. During training, we monitored the training process via the Weights & Biases.

### 8.1.2 Public Benchmark

We performed our experiments on four public benchmarks: Sports, Beauty, Toys, and Yelp[5]. The Sports, Beauty, and Toys datasets are subcategories of the Amazon Review Dataset [71]. The Sports dataset contains reviews for sporting goods, the Beauty dataset contains reviews for beauty products, and the Toys dataset contains toy reviews. On the other hand, the Yelp dataset focuses on business recommendations and is provided by Yelp company. Table 6 summarizes the datasets' details. We only kept datasets where all users and items have at least five interactions. Besides, we adopted the dataset split settings used in the previous method [18].

### 8.1.3 Evaluation Metric

To evaluate ELCRec, we adopt two groups of metrics, including Hit Ratio@$k$ (HR@$k$) and Normalized Discounted Cumulative Gain@$k$ (NDCG@$k$), where $k \in \{5, 20\}$.

### 8.1.4 Compared Baseline

We compare our method with nine baselines including BPR-MF [88], GRU4Rec [34], Caser [96], SASRec [39], DSSRec [69], BERT4Rec [94], S3-Rec [129], CL4SRec [108], and ICLRec [18]. Detailed introductions to these methods are in the Appendix 8.11.2.

---

[5]https://www.yelp.com/dataset

### 8.1.5 Implementation Detail

For the baselines, we adopt their original code with the original settings to reproduce the results on four benchmarks. Due to page limitation, the detailed implementation of the baselines are listed in Appendix 8.12. The proposed method, ELCRec, was implemented using the PyTorch deep learning platform. In the Transformer encoder, we employed self-attention blocks with two attention heads. The latent dimension, denoted as $d$, was set to 64, and the maximum sequence length, denoted as $T$, was set to 50. We utilized the Adam optimizer with a learning rate of 1e-3. The decay rate for the first moment estimate was set to 0.9, and the decay rate for the second moment estimate was set to 0.999. The cluster number, denoted as $k$, was set to 256 for the Yelp and Beauty datasets and 512 for the Sports and Toys datasets. The trade-off hyper-parameter, denoted as $\alpha$, was set to 1 for the Sports and Toys datasets, 0.1 for the Yelp dataset, and 10 for the Beauty dataset. During training, we monitored the training process via the Weights & Biases.

## 8.2 Notation and Dataset

We list the basic notations in Table 5. And Table 6 summarizes the datasets' details.

Table 5: Basic notations.

| Notation | Meaning |
|---|---|
| $\mathcal{U}$ | User set |
| $\mathcal{V}$ | Item set |
| $\{S^u\}_{u=1}^{|\mathcal{U}|}$ | Users' behavior sequence set |
| $(S^u)^{v_k}$ | Users' behavior sequence set in view $k$ |
| $d'$ | Dimension number of latent features |
| $d$ | Dimension number of aggregated latent features |
| $b$ | Batch size |
| $k$ | Cluster number |
| $T$ | Maximum sequence length |
| $\mathcal{L}_{\text{cluster}}$ | Clustering loss |
| $\mathcal{L}_{\text{seq\_cl}}$ | Behavior sequence contrastive loss |
| $\mathcal{L}_{\text{intent\_cl}}$ | Intent contrastive loss |
| $\mathcal{L}_{\text{icl}}$ | intent-assisted contrastive learning loss |
| $\mathcal{L}_{\text{next\_item}}$ | Next item prediction loss |
| $\mathcal{L}_{\text{overall}}$ | Overall loss of the proposed ELCRec |
| $\mathcal{F}$ | Behavior Encoder |
| $\mathcal{P}$ | Concatenate pooling function |
| $\mathbf{E}^u \in \mathbb{R}^{|S^u| \times d'}$ | Behavior sequence embedding of user $u$ |
| $\mathbf{H} \in \mathbb{R}^{|\mathcal{U}| \times d}$ | Behavior embeddings of all users |
| $\hat{\mathbf{H}} \in \mathbb{R}^{|\mathcal{U}| \times d}$ | Normalized Behavior embeddings of all users |
| $\mathbf{H}^{v_k} \in \mathbb{R}^{|\mathcal{U}| \times d}$ | Behavior embeddings of all users in view $v_k$ |
| $\mathbf{C} \in \mathbb{R}^{k \times d}$ | Learnable cluster center embeddings |
| $\hat{\mathbf{C}} \in \mathbb{R}^{k \times d}$ | Normalized Learnable cluster center embeddings |

## 8.3 Algorithm Table

We summarize the overall process of the ELCRec method in Algorithm 1.

Table 6: Statistical information of four public datasets.

| Dataset | #User | #Item | #Action | Avg. Len. | Sparsity |
|---|---|---|---|---|---|
| Sports | 35,598 | 18,357 | 0.3M | 8.3 | 99.95% |
| Beauty | 22,363 | 12,101 | 0.2M | 8.9 | 99.95% |
| Toys | 19,412 | 11,924 | 0.17M | 8.6 | 99.93% |
| Yelp | 30,431 | 20,033 | 0.3M | 8.3 | 99.95% |

---

**Algorithm 1** End-to-end Learnable Clustering Framework for Recommendation (ELCRec)

---

**Input**: user set $\mathcal{U}$; item set $\mathcal{V}$; historical behavior sequences $\{\mathcal{S}^u\}_{u=1}^{|\mathcal{U}|}$; cluster number $k$; epoch number $E$; learning rate; trade-off parameter $\alpha$.
**Output**: Trained ELCRec.

1: Initialize model parameters in encoders.
2: **for** epoch $= 1, 2, ..., E$ **do**
3:     **for** u $= 1, 2, ..., |\mathcal{U}|$ **do**
4:         Obtain $u$-th user's behavior sequence embedding $\mathbf{E}^u \in \mathbb{R}^{|\mathcal{S}^u| \times d'}$ via encoding $\mathcal{S}^u$ in Eq. (1).
5:         Obtain $u$-th user's aggregated behavior embedding $\mathbf{h}_u \in \mathbb{R}^{1 \times d}$ via aggregating $\mathbf{E}^u$ in Eq. (2)
6:     **end for**
7:     Obtain behavior embeddings of all users $\mathbf{H} \in \mathbb{R}^{|\mathcal{U}| \times d}$.
8:     Initialize cluster centers $\mathbf{C} \in \mathbb{R}^{k \times d}$ as learnable.
9:     Calculate clustering loss to conduct intent learning.
10:     Generate two views of behaviors via data augmentations.
11:     Encode the two views of the behavior sequences.
12:     Calculate $\mathcal{L}_{\text{seq\_cl}}$ to conduct behavior contrastive learning.
13:     Query cluster index of the behavior embeddings via Eq. (5).
14:     Fuse the intent information to behavior embeddings.
15:     Calculate $\mathcal{L}_{\text{intent\_cl}}$ to conduct intent contrastive learning.
16:     Calculate $\mathcal{L}_{\text{next\_item}}$ to conduct next item prediction task.
17:     Minimize $\mathcal{L}_{\text{overall}}$ to train the model in Eq. (8).
18: **end for**
19: **Return** Well-trained ELCRec model.

---

## 8.4 Applicability on Diverse Domains

To further demonstrate the applicability of ELCRec on different recommendation domains, we conduct additional experiments on movie recommendation and news recommendation.

### 8.4.1 Movie Recommendation

For the movie recommendation, we conducted experiments on the MovieLens 1M dataset (ML-1M) [29]. This dataset contains 1M ratings from about 6K users on about 4K movies, as shown in Table 7. In this experiment, we compared our proposed ELCRec with the most related baseline ICLRec. The experimental results are presented in the Table 8.

Table 7: Statistical information of ML-1M dataset.

| Dataset | #User | #Movie | #Rating | Rating per User | Rating per Movie |
|---|---|---|---|---|---|
| **ML-1M** | 6,040 | 3,706 | 1,000,209 | 166 | 270 |

From these experimental results, we draw two conclusions as follows.

Table 8: Recommendation performance on ML-1M dataset. **Bold values** denote the best results. * indicates the $p$-value<0.05.

| Method | HR@5 | HR@20 | NDCG@5 | NDCG@20 |
|--------|------|-------|--------|---------|
| ICLRec | 0.0293 | 0.0777 | 0.0186 | 0.0320 |
| ELCRec | **0.0333** | **0.0836** | **0.0208** | **0.0347** |
| Impro. | 13.65%↑ | 7.59%↑ | 11.83%↑ | 8.44%↑ |
| $p$-value | 4.03e-6* | 6.68e-9* | 6.36e-6* | 1.66e-6* |

(a) ELCRec achieves better recommendation performance, as evidenced by higher values for all four metrics: HR@5, HR@20, NDCG@5, and NDCG@20. For example, with the HR@5 metric, ELCRec outperforms ICLRec by 13.65%.

(b) We calculated the $p$-value between our method and the runner-up. The results indicate that all the $p$-values are less than 0.05, suggesting that our ELCRec significantly outperforms ICLRec.

(c) We demonstrate the applicability and superiority of the proposed ELCRec in the movie recommendation domain.

### 8.4.2  News Recommendation

In addition, for news recommendation, we aim to conduct experiments on the MIND-small dataset [106]. MIND contains about 160k English news articles and more than 15 million impression logs generated by 1 million users. Every news article contains rich textual content, including title, abstract, body, category, and entities. Each impression log contains the click events, non-clicked events, and historical news click behaviors of this user before this impression. To protect user privacy, each user was de-linked from the production system when securely hashed into an anonymized ID. MIND-small is a small version of the MIND dataset by randomly sampling 50,000 users and their behavior logs from the MIND dataset. We list the experimental results in Table 9.

Table 9: Recommendation performance on MIND-small dataset. **Bold values** denote the best results. * indicates the $p$-value<0.05.

| Method | HR@5 | HR@20 | NDCG@5 | NDCG@20 |
|--------|------|-------|--------|---------|
| ICLRec | 0.0890 | 0.2128 | 0.0578 | 0.0926 |
| ELCRec | **0.0944** | **0.2332** | **0.0603** | **0.0994** |
| Impro. | 6.07%↑ | 9.59%↑ | 4.33%↑ | 7.34%↑ |
| $p$-value | 7.09e-17* | 9.57e-09* | 6.11e-7* | 1.09e-7* |

From these experimental results, we have three conclusions as follows.

(a) ELCRec surpasses the runner-up for all four metrics, including HR@5, HR@20, NDCG@5, and NDCG@20. Significantly, ELCRec improve the runner-up by 9.59% with HR@20.

(b) We conduct $t$-test for ELCRec and the runner-up method and find all the $p$-values are less than 0.05. It indicates that our method significantly outperforms the runner-up method.

(c) We demonstrate the applicability and superiority of the proposed ELCRec in the news recommendation domain.

Overall, we further demonstrate the applicability of ELCRec in diverse domains from the news and movie aspects.

### 8.5  Precise Data of Ablation Study

Due to the limitation of the main pages of the paper, we provide the precise data of the ablation studies in this section.

Table 10: The precise date of the ablation studies. "B", "B+ICL", "B+ELCM", and "ELCRec" denote the baseline, the baseline with intent-assisted contrastive learning, the baseline with the end-to-end learnable clustering module, and the baseline with both, respectively.

|        | B      | B+ICL  | B+ELCM | ELCRec |
|--------|--------|--------|--------|--------|
| Sports | 0.1343 | 0.1379 | 0.1396 | 0.1405 |
| Beauty | 0.239  | 0.2398 | 0.2432 | 0.2473 |
| Toys   | 0.2664 | 0.2675 | 0.2718 | 0.2686 |
| Yelp   | 0.1258 | 0.1262 | 0.1285 | 0.1305 |

## 8.6 Sensitivity

This section aims to answer the research question (iv). To verify the sensitivity of the proposed EL-CRec to hyper-parameters, we test the performance on four datasets with different hyper-parameters. The experimental results are demonstrated in Figure 2. The x-axis denotes the values of hyper-parameters, and the y-axis denotes the values of the HR@5 metric. We obtain two conclusions as follows.

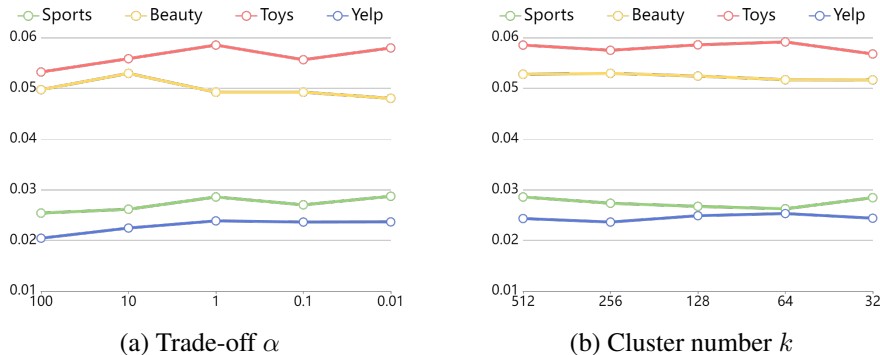

(a) Trade-off $\alpha$          (b) Cluster number $k$

Figure 2: Sensitivity analyses of ELCRec. The results are evaluated by the HR@5 metric.

(a) For the trade hyper-parameter $\alpha$, we test the performance with $\alpha \in \{0.01, 0.1, 1, 10, 100\}$. We find that our proposed ELCRec is not very sensitive to trade-off $\alpha$. And ELCRec can achieve promising performance when $\alpha \in [0.1, 10]$.

(b) For the cluster number $k$, we test the recommendation performance with $\alpha \in \{32, 64, 128, 256, 512\}$. The results show that ELCRec is also not very sensitive to cluster number $k$ and can perform well when $k \in [256, 512]$.

## 8.7 Convergence

To answer the research question (v), we monitor the recommendation performance and training loss as shown in Figure 3. We find that the losses gradually decrease and eventually converge. Besides, during the training process, the recommendation performance gradually increases and eventually reaches a promising value.

## 8.8 Visualization

We conduct visualization experiments on four public datasets to further demonstrate ELCRec's capability to capture users' underlying intents. Concretely, the learned behavior embeddings are visualized via $t$-SNE during training. As shown in Figure 6, the first row to the fourth row denotes the results on Sports, Beauty, Toys, and Yelp, respectively. From these experimental results, we have three observations as follows.

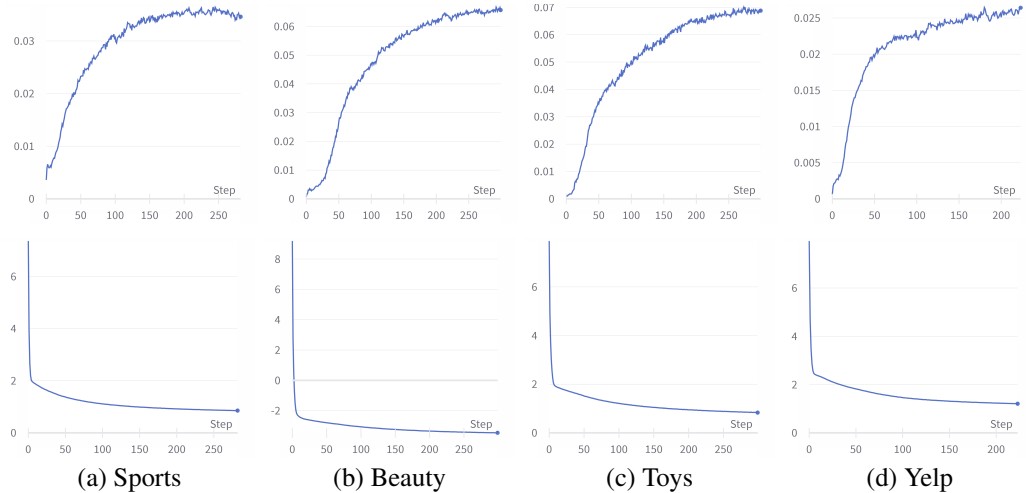

|  | (a) Sports | (b) Beauty | (c) Toys | (d) Yelp |

Figure 3: Convergence analyses. The first and second row denotes HR@5 on the evaluation set and training loss, respectively.

## 8.9    Additional Cost Experiment

We provide the additional cost experiments in this section. Concretely, we add the conventional self-supervised-learning-based sequential recommendation method S3-Rec in the cost comparison experimens, since ICLRec is based on S3-Rec and comparing other regular methods is not very informative. The experimental results are demonstrated as follows. We find that the conventional self-supervised-learning-based recommendation method S3-Rec costs more time and memory compared with the ICLRec and ELCRec since 1) it contains two training phases, including the pre-training and the fine-tuning. 2) It incorporates four complex self-supervised learning tasks, including associated attribute prediction, masked item prediction, segment prediction, and masked item prediction.

Table 11: Running time and memory costs.

| Cost | Dataset | Sports | Beauty | Toys | Yelp | Average |
|------|---------|--------|--------|------|------|---------|
|        | S3-Rec | 8319 | 4414 | 4452 | 5925 | 5778 |
| **Time** | ICLRec | 5282 | 3770 | 4374 | 4412 | 4460 |
|        | ELCRec | 5360 | 2922 | 4124 | 4151 | 4139 |
|        | S3-Rec | 2512 | 2294 | 2975 | 3982 | 2941 |
| **Memory** | ICLRec | 1944 | 1798 | 2887 | 3671 | 2575 |
|        | ELCRec | 1781 | 1574 | 2555 | 3383 | 2328 |

## 8.10    Practical Insights

In this section, we provide practical experiences and insights for the deployment of our proposed method. They contain three parts, including a case study, solutions to rapid shift problem, and solutions to balance problems.

### 8.10.1    Case Study

To explore how our proposed method works well, we conduct case studies on large-scale industrial data. They contain two parts: case studies on user group distribution and case studies on the learned clusters.

Firstly, for the user group distribution, the results are demonstrated in Figure 4. We visualize the cluster distribution of different user groups. "top" denotes the cluster IDs that have the highest proportion in the user group. "bottom" denotes the cluster IDs that have the lowest proportion in the user group. From these analyses, we have two findings as follows.

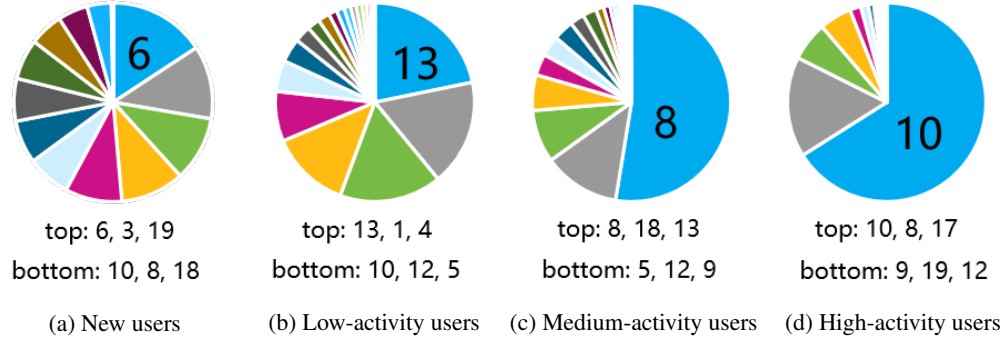

| (a) New users | (b) Low-activity users | (c) Medium-activity users | (d) High-activity users |

Figure 4: Case studies on different user groups. The distributions of different user groups are visualized. "top" denotes the cluster IDs, which have the highest proportion in the user group. "bottom" denotes the cluster IDs, which have the lowest proportion in the user group.

(a) As the user activity increases, the distribution becomes sharper. Namely, the users who have higher activities tend to distribute to one or two clusters. For example, about 60% of the high-activity users are attributed to cluster 10.

(b) The "top" cluster IDs of the high-activity user group, such as cluster 10 and cluster 8, are exactly the "bottom" cluster IDs of the low-activity user group. Similarly, the "bottom" cluster IDs of the high-activity user group, such as cluster 9, are exactly the "top" cluster IDs of the low-activity user group. This indicates that the learned cluster centers can well separate different user groups.

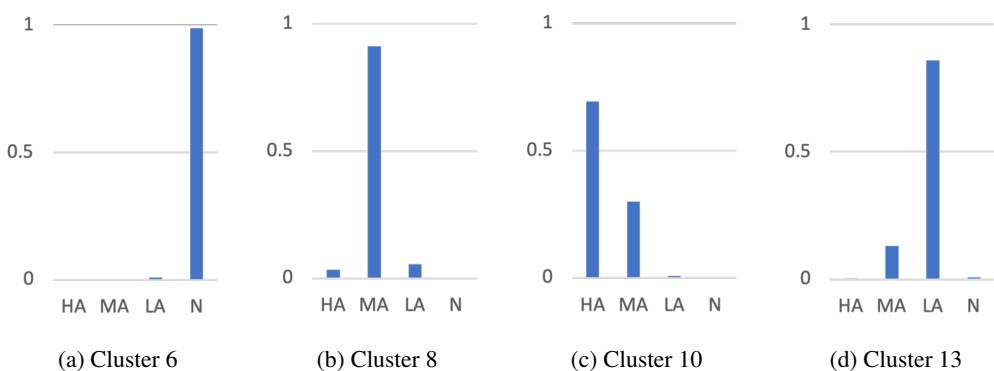

| (a) Cluster 6 | (b) Cluster 8 | (c) Cluster 10 | (d) Cluster 13 |

Figure 5: Case studies on the learned cluster. We visualize the distribution of the learned clusters. "HA", "MA", "LA", and "N" denote the high-activity, medium-activity, low-activity, and new user groups, respectively.

Secondly, we also conduct extensive case studies on the learned clusters. To be specific, we analyze the user distribution of each cluster, as shown in Figure 5. From the results, we can observe that, in cluster 6, most users are new. Besides, in the cluster 8, the most users are with medium activity. In addition, in cluster 10, most users are with high activity and medium activity. Moreover, in cluster 13, most users are with low activity and medium activity. Previous observations show that the learned centers can separate the users into different groups based on their activities.

In summary, these case studies further verify the effectiveness of ELCRec. Also, they provide insights for future work.

### 8.10.2 Solutions to Rapid Shift Problem

On real-time large-scale industrial data, the users' behaviors and intents will shift rapidly. Therefore, we argue that the existing EM optimization can not capture the latest users' intents, thus easily misunderstanding users and harming recommendation performance. Fortunately, our proposed ELCRec method can alleviate this problem. Concretely, the end-to-end learnable cluster module can guide the network to learn users' intents dynamically. It can update the learned clusters (intents) at each batch, satisfying the requirement of rapid update. However, our method makes it hard to control the update rate of the users' intents. That is one of drawbacks of ELCRec, we will discuses it and the potential solution in 8.14.

### 8.10.3 Solutions to Balance Problem

Balancing the different loss functions in our model is indeed an important challenge. Our overall loss function consists of next-item prediction loss, intent-assisted contrastive loss, and cluster loss. It is formulated as follows: $\mathcal{L}_{\text{overall}} = \mathcal{L}_{\text{next\_item}} + 0.1 \times \mathcal{L}_{\text{icl}} + \alpha \times \mathcal{L}_{\text{cluster}}$. We set the weight of $\mathcal{L}_{\text{icl}}$ as 0.1 to maintain it in the same order of magnitude as the first term. This reduces the number of hyper-parameters and simplifies the selection process. The weight of $\mathcal{L}_{\text{cluster}}$ is set as a hyper-parameter $\alpha$. We test different values of $\alpha \in \{0.01, 0.1, 1, 10, 100\}$ and find that our ELCRec method is not very sensitive to the trade-off $\alpha$. Promising performance is achieved when $\alpha \in [0.1, 10]$. The sensitivity analysis experiments are presented in Figure 2 (b). In our proposed model, we set $\alpha$ to 1 for the Sports and Toys datasets, 0.1 for the Yelp dataset, and 10 for the Beauty dataset. The selection of $\alpha$ is mainly based on the model performance. We provide several practical strategies to balance multiple losses in multi-task learning.

- Weighted Balancing. Assign weights to each loss function to control their contribution to the overall loss. By adjusting the weights, a balance can be achieved between different loss functions. This can be determined through prior knowledge, empirical rules, or methods like cross-validation.

- Dynamic Weight Adjustment. Adjust the weights of the loss functions in real time based on the model's training progress or the characteristics of the data. For example, dynamically adjust the weights based on the model's performance on a validation set, giving relatively smaller weights to underperforming loss functions.

- Multi-objective Optimization. Treat different loss functions as multiple optimization objectives and use multi-objective optimization algorithms to balance these objectives. This allows for the simultaneous optimization of multiple loss functions and seeks balance between them.

- Gradient-based Adaptive Adjustment. Adaptively adjust the weights of loss functions based on their gradients. If a loss function has a larger gradient, it may have a greater impact on the model's training, and its weight can be increased accordingly.

- Ensemble Methods. Train multiple models based on different loss functions and use ensemble learning techniques to combine their prediction results. By combining the predictions of different models, a balance between different loss functions can be achieved.

In the future, we will continue to improve our model based on the above strategies.

(a) At the beginning of training, the behavior embeddings are disorganized and can not reveal the underlying intents.

(b) During the training process, the latent distribution is optimized, and similar behaviors are grouped into latent intents.

(c) After optimization, the users' underlying intents appear, and we highlight them with circles in Figure 6. These intents can assist recommendation systems in better modeling users' behavior and recommending items. In summary, the above experiments and observations verify the effectiveness of our proposed ELCRec.

## 8.11 Detailed Related Work

### 8.11.1 Sequential Recommendation

Sequential Recommendation (SR) poses a significant challenge as it strives to accurately capture users' evolving interests and recommend relevant items by learning from their historical behavior

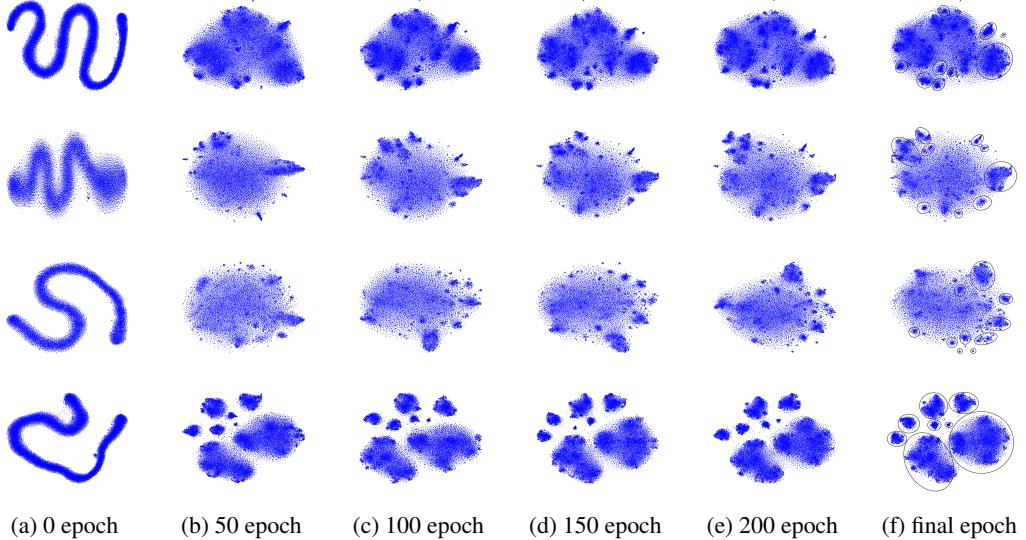

|   |   |   |   |   |   |
|---|---|---|---|---|---|
| (a) 0 epoch | (b) 50 epoch | (c) 100 epoch | (d) 150 epoch | (e) 200 epoch | (f) final epoch |

Figure 6: *t*-SNE visualization on four public datasets. The first row to the fourth row denotes the results on Sports, Beauty, Toys, and Yelp.

sequences. In the early stages, classical techniques such as Markov Chains and matrix factorization have assisted models [32, 86, 87] in learning patterns from past transactions. Deep learning has garnered significant attention in recent years and has demonstrated promising advancements across various domains, including vision and language. Inspired by the remarkable success of Deep Neural Networks (DNNs), researchers have developed a range of deep Sequential Recommendation (SR) methods. For instance, Caser [96] leverages Convolutional Neural Networks (CNNs) [42] to embed item sequences into an "image" representation over time, enabling the learning of sequential patterns through convolutional filters. Similarly, GRU4Rec [34] utilizes Recurrent Neural Networks (RNNs) [121], specifically the Gated Recurrent Unit (GRU), to model entire user sessions. The Transformer architecture [100] has also gained significant popularity and has been extended to the recommendation domain. For example, SASRec [39] employs a unidirectional Transformer to model users' behavior sequences, while BERT4Rec [94] utilizes a bidirectional Transformer to encode behavior sequences from both directions. To enhance the time and memory efficiency of Transformer-based SR models, LSAN [50] introduces aggressive compression techniques for the original embedding matrix. Addressing the cold-start issue in SR models, ASReP [66] proposes a pre-training and fine-tuning framework. Furthermore, researchers have explored the layer-wise disentanglement of architectures [126] and introduced novel modules like the Wasserstein self-attention module in STOSA [25] to model item-item position-wise relationships. In addition to Transformers, graph neural networks [117, 125, 52, 17] and multilayer perceptrons [48, 47, 130] have also found applications in recommendation systems. More recently, Self-Supervised Learning (SSL) [119, 84], particularly contrastive learning [38], has gained popularity due to its ability to learn patterns from large-scale unlabeled data. As a result, SSL-based SR models have been increasingly introduced. For instance, in CoSeRec [65], Liu et al. propose two informative augmentation operators that leverage item correlations to generate high-quality views. They then utilize contrastive learning to bring positive sample pairs closer while pushing negative pairs apart. Subsequently, TiCoSeRec [20] is designed by considering the time intervals in the behavior sequences. Another contrastive SR method, ECL-SR [131], ensures that the learned embeddings are sensitive to invasive augmentations while remaining insensitive to mild augmentations. Additionally, DCRec [116] addresses the issue of popularity bias through a debiased contrastive learning framework. Moreover, DuoRec [82] is proposed to solve the representation degeneration problem in contrastive recommendation methods. Techniques such as hard negative mining [24, 78] have also proven beneficial for recommendation systems. Besides, motivated by the success of Mask Autoencoder (MAE) [31], MAERec [118] is proposed with the graph masked autoencoder.

### 8.11.2 Intent Learning for Recommendation

The preferences of users towards items are implicitly reflected in their intents. Recent studies [44, 14, 45, 18, 49, 53, 5] have highlighted the significance of users' intents in the user understanding and enhancing the performance of recommendation systems. For instance, MCPRN [103] introduces a mixture-channel method to model subsets of items with multiple purposes. Inspired by capsule networks [92], MIND [44] utilizes dynamic routing to learn users' multiple interests. Furthermore, ComiRec [14] employs a multi-interest module to capture diverse interests from user behavior sequences, while the aggregation module combines items from different interests to generate overall recommendations. Besides, MITGNN [64] treats intents as translated tail entities and learns embeddings using graph neural networks. In addition, Pan et al. [77] propose an intent-guided neighbor detector to identify relevant neighbors, followed by a gated fusion layer that adaptively combines the current session with the neighbor sessions. Moreover, Ma et al. [69] aim to disentangle the intentions underlying users' behaviors and construct sample pairs within the same intention. Meanwhile, the ASLI method [97] incorporates a temporal convolutional network layer to extract latent users' intents. More recently, a general latent learning framework called ICLRec [18] is introduced, which utilizes contrastive learning [127, 128] and $k$-Means clustering to group the users' behaviors to intents. Chang et al. [15] formulate users' intents as latent variables and infer them based on user behavior signals using the Variational Auto Encoder (VAE) [40]. To mitigate noise caused by data augmentations in contrastive SR models, IOCRec [49] proposes building high-quality views at the intent level. Besides, ICSRec [81] is proposed to solve this issue by conducting contrastive learning on cross sub-sequences. DIMPS [5] aims to build dynamic and intent-oriented document representations for intent learning. PoMRec [22] insert the specific prompts into user interactions to make them adaptive to different learning objectives. Furthermore, Teddy [53] is proposed by utilizing the intent trend and diversity.

Firstly, we want to clearly claim the target of this paper and the demand of the industrial scenario as follows. 1) For the open benchmarks, we aim to develop an intent learning method to decouple user's intents for a better recommendation based on the appropriate intents of the user. 2) For the industrial data, we aim to design a user grouping method to cluster the users into different groups to solve the cold-start problem via mapping the new users into the user group, which contains more useful information. Therefore, the designed method needs to have the following abilities. 1) It can explicitly decouple users' behaviors into different intents (grouping users into different clusters). 2) It can be easily adapted to large-scale real-time industrial data, saving memory and time costs. Secondly, we surveyed massive recent state-of-the-art methods to solve the above challenges in the related work part of this paper. We highlight the drawbacks of the related methods [49] [3] and claim why they will fail in our scenario. In the IOCRec method [49], they define the prototype intention of users as a $k \times d$ matrix. And the these prototype intention are directly used to calculate the relevance weights and the intentions. However, there are no designs for the initialization and optimization of the prototype intention, e.g., guiding the prototype intention to represent the users' behaviors, and different intentions are separated. Therefore, it lacks explainability and persuasiveness, especially in the scenario where there is a need to conduct different recommendation strategies for different groups, i.e., user grouping recommendations. Also, we do not find theoretical or experimental evidence to support that the learned intents are separated well and reveal the representative behaviors of users in the original paper [49]. For the DCCF method [85], 1) it is based on the graph neural networks, limiting the model scalability and efficiency on large-scale data due to the large costs of graph constructing, graph storage, and neighbor sampling. And the sequential methods are more efficient since our data is naturally the sequences of the user behaivors. 2) Besides, in the DCCF method, the intents are randomly initialized via Xavier normalization. Then, they are used to aggregate information. In the loss function part, we notice that there is only a penalty item to limit the complexity of the parameters of intent embeddings. Thus, there are no operations or loss functions to explicitly optimize the users' intents, such as separating different intents, learning intents from behaviors, etc. We claim this intent decoupling is relatively weak and may not really learn well and separate the different intents of users. Also, in Figure 4 of the original paper [85], we find that the cluster pattern is not revealed well in the sampled data. We speculate the cluster pattern will also not be revealed well on the whole samples of the datasets. Thirdly, we explain why we chose ICLRec [18] as our baseline. 1) ICLRec is a sequential recommendation method, which is more suitable for our data. Compared to the GNN-based methods, it can save more time and memory costs. 2) ICLRec adopts the clustering algorithm to explicitly separate the users' intents, which can also be adapted for user grouping. It explicitly optimizes the intents based on the users' behavior embeddings. We believe this technique can better seperate the users' intents well and also better

obtain the users' intents from their behaviors. In Figure 7 of the original paper, [18], we find that ICLRec can reveal the cluster pattern well on the sampled data. Fourthly, we claim our motivation. Although ICLRec can achieve promising performance and effectively decouple users' intents, the EM optimization framework limits the scalability and performance. 1) At the E-step, we need to apply the clustering algorithm on the whole data, limiting the model's scalability, especially in large-scale industrial scenarios, e.g., apps with billion users. 2) In the EM framework, the optimization of behavior learning and the clustering algorithm are separated, leading to sub-optimal performance and increasing the implementation difficulty. We admit that our analyses of the problems start from ICLRec methods. But, actually, there are many intent learning methods [81, 70, 72, 74, 98] that adopt the clustering algorithms and the EM framework. They will meet the above problems and may fail when scaling to real-time large-scale data. Therefore, we claim our mentioned challenges are general recommendation systems, especially for intent decoupling methods. We believe our proposed end-to-end learnable clustering module can bring performance improvement and save time and space costs for these methods.

### 8.11.3    VQ/RQ-based Recommendation

VQ-Rec [35] is proposed to solve the issues, including over-emphasizing the effect of text features and exaggerating the negative impact of the domain gap by learning the vector-quantized item representation. The schema of VQ-Rec is summarized as text->code->representation. However, VQ-Rec mainly focuses on item representation, and the number of items is always largely smaller than the number of users in the large-scale recommenders. In addition, the original paper of VQ-Rec mentions that "the used technique for training OPQ, i.e., k-means, tends to generate clusters with a relatively uniform distribution on ...". It seems that VQ-Rec adopts the conventional k-means clustering for the code; therefore may lead to out-of-memory and long training time problems. Similarly, [37] proposes an extremely memory-efficient factorization machine named xLightFM, where each category embedding is composited with latent vectors selected from the codebooks. xLightLM is a factorization-machine-based recommendation method, which is different from the sequential recommendation methods and makes it hard to process the sequence data. Additionally, in the original paper of xLightLM, the authors mentioned: "..., which first decomposes the embedding space into the Cartesian product of subspaces and conducts the k-means clustering in each subspace for obtaining center vectors". It also simply adopts the k-means clustering algorithm on the embedding to obtain the codebooks. Thus, it also meets the out-of-memory and long training time problems on large-scale data. Moreover, a generative retrieval approach named TIGER [83] is proposed by creating a semantically meaningful tuple of codewords to serve as a Semantic ID for each item. Although the residual quantization is verified effective, the method seems still based on offline clustering since the authors mentioned, "we use k-means clustering-based initialization for the codebook." In addition, it also mainly focuses on the item embeddings and aims to provide the semantical information for the items. Different from them, our method mainly focuses on the user embeddings, which are more numerous compared with the items. Also, our proposed method utilizes end-to-end learnable clustering to unify intent learning and behavior learning in a unified framework. It not only improves the recommendation performance but also improves the scalability of the intent learning method. The evidence can be found in the experiment part of the paper. Moreover, these three related papers seem not to focus on the intent learning of users.

### 8.11.4    Clustering Algorithm

Clustering is a fundamental and challenging task that aims to group samples into distinct clusters without supervision. By leveraging the power of unlabeled data, clustering algorithms have found applications in various domains, including computer vision [16, 7], natural language processing [3], graph learning [61, 9, 115], and recommendation systems [18, 81, 112, 114]. In the early stages, several traditional clustering methods [30, 101, 89, 23, 90, 120] were proposed. For instance, the classical $k$-Means clustering [30] iteratively updates cluster centers and assignments to group samples. Spectral clustering [101] constructs a similarity graph and utilizes eigenvalues and eigenvectors to perform clustering. Additionally, probability-based Gaussian Mixture Models (GMM) [89] assume that the data distribution is a mixture of Gaussian distributions and estimate parameters through maximum likelihood. Moreover, the repulsive clustering methods [43, 21, 2] cluster data via the repulsive terms. In contrast, density-based methods [23, 90, 19] overcome the need for specifying the number of clusters as a hyperparameter. In recent years, the impressive performance of deep

learning has sparked a growing interest in deep clustering [51, 91, 73, 4, 80, 46, 27, 26, 8, 10, 104]. For instance, Xie et al. propose DEC [107], a deep learning-based approach for clustering. They initialize cluster centers using $k$-Means clustering and optimize the clustering distribution using a Kullback-Leibler divergence clustering loss [107]. IDEC [28] improves upon DEC by incorporating the reconstruction of original information from latent embeddings. JULE [109] and DeepCluster [11] both adopt an iterative approach, updating the deep network based on learned data embeddings and clustering assignments. SwAV [12], an online method, focuses on clustering data and maintaining consistency between cluster assignments from different views of the same image. DINO [13] introduces a momentum encoder to address representation collapse. Additionally, SeCu [79] proposes a stable cluster discrimination task and a hardness-aware clustering criterion. While deep clustering has been extensively applied to image data, it is also utilized in graph clustering [57, 63, 102, 59, 58, 76, 61, 62, 60, 110, 111, 113] and text clustering [3, 56, 36, 93]. However, the application of clustering-based recommendation [18, 81] is relatively unexplored. Leveraging the unsupervised learning capabilities of clustering could benefit intent learning in recommendation systems.

## 8.12  Implementation Details of Baselines

For the baseline methods, we adopt the public source code with the default parameter settings and reproduce their results on the four benchmarks. The source codes of these methods are available in Table 12. Besides, for the used benchmarks, following [18], we only kept datasets where all users and items have at least five interactions. Besides, we adopted the dataset split settings used in [18]. The Sports, Beauty, and Toys datasets [71, 33] are obtained from: http://jmcauley.ucsd.edu/data/amazon/index.html. The Yelp dataset is obtained from https://www.yelp.com/dataset.

For the results that have already existed in the original papers, we reuse them in our paper. For the results that do not exist in the original papers, we adopt the official codes of the baselines to reproduce the experimental results. Concretely, for the hyperparameters, we adopt and try several sets of the default hyperparameters on different datasets released by the original authors. We report the best result obtained from the best hyper-parameters. By the way, we also observe these results have already converged well. Besides, we conducted three runs on different random seeds for all experimental results and reported the average performance.

Table 12: Implementation URLs of baseline methods.

| Method | Url |
| --- | --- |
| BPR-MF [88] | https://github.com/xiangwang1223/neural_graph_collaborative_filtering |
| GRU4Rec [34] | https://github.com/slientGe/Sequential_Recommendation_Tensorflow |
| Caser [96] | https://github.com/graytowne/caser_pytorch |
| SASRec [39] | https://github.com/kang205/SASRec |
| BERT4Rec [94] | https://github.com/FeiSun/BERT4Rec |
| DSSRec [69] | https://github.com/abinashsinha330/DSSRec |
| S3-Rec [129] | https://github.com/RUCAIBox/CIKM2020-S3Rec |
| CL4SRec [108] | https://github.com/HKUDS/SSLRec |
| ICLRec [18] | https://github.com/salesforce/ICLRec |
| DCRec [116] | https://github.com/HKUDS/DCRec |
| MAERec [118] | https://github.com/HKUDS/MAERec |
| IOCRec [49] | https://github.com/LFM-bot/IOCRec |

## 8.13  Deployment Details

We aim to apply our proposed method to the real-time large-scale industrial recommendation systems. Concretely, the ELCRec algorithm is applied to live streaming recommendations on the front page of the Alipay app. The user view (UV) and page view (PV) of this application are about 50 million and 130 million, respectively. Since most of the users are new to this application, it easily leads to the sparsity of users' behaviors, namely, the cold-start problem in recommendation systems. Our proposed ELCRec can alleviate this problem by grouping users and then making recommendations.

This method can map a new user to a user group, which contains more intent behaviour information from similar users, such as other similar new users and similar users with low/middle activities. In this manner, we can guide the model to learn the behavior of new users and provide more precise recommendations for them even with sparse behaviors.

First, we introduce the online baseline of this project. Since the sparsity of the users' behaviors, we replaced the users' behaviors with the users' activities. Then, the online baseline multi-gate mixture-of-expert (MMOE) [68] models the users' activities. In this model, the experts are designed to extract the features of users, and the multi-gates are designed to select specific experts. The inputs of the multi-gates are the activities of the users. This design aims to train an activity-awarded model to group different users and then conduct recommendations.

However, we found the performance of this model is limited, and the output of the gates is smooth, indicating that this model may fail to group users. Meanwhile, on the open benchmarks, extensive experiments demonstrate the proposed end-to-end learnable clustering module is effective and scalable. Thus, to solve the above problem, ELCRec is adopted in this project. It is designed to assist the gate to group users. For example, the high-activity users and new users are grouped into different clusters, and then the users in different groups will be recommended differently. Therefore, it alleviates the cold-start issue and further improves the recommendation performance. Besides, during the learning process of the cluster embeddings, the low-activity users can transfer to high-activity users, improving the overall users' activities in the application. It is worth mentioning that the networks are trained with multi-task targets, e.g., CTR prediction, CVR prediction, etc. Following the previous online baseline, the method is implemented with the TensorFlow deep learning platform [1].

We discuss the user group assignment problem at two different stages of the recommendation. For the recommendation produced by the model, i.e., at the rank stage, it just needs to separate the different user groups and provide personalized recommendations for new users and users with high activities, and it does not need to know which groups are exactly the new user group or the high-activity user group. This way can already provide personalized recommendations for different user groups and solve the cold-start problem in recommendation. Moreover, at the pre-rank stage, we may design some recommendation strategies for different user groups. Therefore, we need to know the clustering assignment of the different user groups. Note that, after training and clustering, we can obtain the clustering assignment of all samples (users). And then we need to label the different user groups based on the user activities or other manual tags of the users by some simple strategies, such as voting and ensemble. After labeling different user groups, we can provide different recommendation strategies, such as boosting or un-boosting for different user groups. In summary, at the rank stage, there is no need for the model inference to provide the exact labels for each user group. Besides, at the re-rank stage, if we want to design some strategies for different user groups, we can adopt the vote or ensemble methods to label the user group embeddings based on their activities or other manual tags of the users.

## 8.14 Limitations & Future Work

In this paper, we propose a novel intent learning method named ELCRec based on the end-to-end learnable clustering framework. It can better mine users' underlying intents via unifying representation learning and clustering optimization. Besides, the end-to-end learnable clustering module optimizes the clustering distribution via mini-batch data, thus improving the scalability and convenience of deployment. Moreover, we demonstrate the superiority, effectiveness, efficiency, sensitivity, convergence, and visualization of ELCRec on four benchmarks. ELCRec is also successfully applied in the real-time large-scale industrial recommendation system. Although achieving promising results, we admit the proposed ELCRec algorithm has several limitations and drawbacks. We summarize them as follows.

- Pre-defined Cluster Number. The cluster number in ELCRec is a pre-defined hyperparameter. In real-time large-scale data, it is hard to determine the cluster number, especially under unsupervised conditions. In this paper, for the open benchmarks, we search the cluster number in {32, 64, 128, 256, 512}. Besides, for the industrial application, the cluster number is set to 20 based on the number of user groups. However, either the search method or the expert knowledge can not determine the cluster number well at once. The cluster number may change dynamically during model training, and the proposed method may fail to achieve promising performance.

- Limited Recommendation Domains. In this paper, we adopt four recommendation benchmarks, including Sports, Beauty, Toys, and Yelp, for the main experimental results. But, these four datasets are all buying recommendation datasets. Besides, we adopt ML-1M [29] and MIND-small [106] for the movie and news recommendation for the additional experiments. However, the recommendation domains are still limited. In the future, we can further demonstrate the boarder applicability of ELCRec in other domains.

- Uncontrollable Update Rate of Cluster Centers. In the real-time recommendation system, the users' behaviors and intents usually change rapidly. Although our proposed ELCRec can dynamically learn the users' intents, it is hard to control the update rate of the underlying clusters (intents).

To solve these issues, we summarize several future works and the potential technical solutions as follows.

- Density-based Clustering. As mentioned above, the cluster number is a pre-defined value in this paper, limiting the recommendation performance and flexibility of the method. To solve this issue in the future, firstly, we can determine the cluster number based on some cluster number estimation methods. They can help to determine the cluster number by performing multiple clustering runs and selecting the best cluster number based on the unsupervised criterion. The mainstream cluster number estimation methods [41] include the thumb rule, ELBOW [95], $t$-SNE [99], etc. The thumb rule simply assigns the cluster number $k$ with $\sqrt{n/2}$, where $n$ is the number of samples. This manual setting is empirical and can not be applicable to all datasets. Besides, the ELBOW is a visual method. Concretely, they start the cluster number $k = 2$ and keep increasing $k$ in each step by 1, calculating the WSS (within-cluster sum of squares) during training. They choose the value of $k$ when the WSS drops dramatically, and after that, it reaches a plateau. However, it will bring large computational costs since the deep neural network needs to be trained with repeated times. Another visual method termed $t$-SNE visualizes the high-dimension data into 2D sample points and helps researchers determine the cluster number. The effectiveness of $t$-SNE heavily relies on the experience of researchers. Therefore, secondly, we can determine the cluster number based on the data density [90, 91]. Concretely, the areas with high data density are identified as the cluster centers, while the areas with low data density are identified as the decision boundaries between cluster centers. Besides, reinforcement learning is also a potential solution [59]. Through these designs, the cluster number will be changeable during the training process. It will be determined based on the embeddings itself, better revealing the users' behavior and may achieve better recommendation performance.

- More Recommendation Domains. As mentioned above, the applied recommendation domains of our method are limited. We aim to test ELCRec on more recommendation domains, such as music recommendation [123, 6], group recommendation [124, 55], group buying [122], bundle recommendation [132], etc.

- Controllable Intent Learning. As mentioned above, in the real-time recommendation system, the intents of the users may change rapidly. Our method makes it hard to control the intent update rate during training and inference. To this end, in the future, we can propose a controllable cluster center learning method, such as the momentum update, to control the change rate of the users' intents. Concretely, $\mathbf{C}_t = m \cdot \mathbf{C}_t + (1 - m) \cdot \mathbf{C}_{t-1}$. Here, $\mathbf{C}_t$ denote the cluster center embeddings at $t$ and $m$ denotes the momentum. Then, the cluster centers (intents of users) will be changed rapidly when $m$ is large, and the cluster centers (intents of users) will be changed slowly when $m$ is small. This strategy will control the change rate of the users' intent embeddings, therefore alleviating the above problem.

