# OpenReview forum: "End-to-end Learnable Clustering for Intent Learning in Recommendation"
_NeurIPS.cc/2024/Conference — NeurIPS 2024 poster_

### Official Review · Reviewer_py4J · 2024-07-11

**Soundness:** 3
**Presentation:** 3
**Contribution:** 3
**Rating:** 4
**Confidence:** 3

**Summary:**

This paper introduces an end-to-end learnable clustering framework for intent learning in recommendation systems, termed ELCRec. The current intent recognition methods can be likened to the Expectation-Maximization (EM) algorithm, where the E-step involves clustering to obtain intents, and the M-step uses self-supervised methods to update embeddings. However, these methods suffer from slow clustering speeds and limited scalability, as well as performance issues due to the separation of clustering and optimization processes. ELCRec addresses these issues by integrating user behavior embeddings into a user embedding and introducing a differentiable clustering method (ELCM) that optimizes clustering and intent alignment. Additionally, the framework employs intent-assisted contrastive learning (ICL) and incorporates a next item prediction loss to enhance the recommendation performance.

**Strengths:**

1.	The proposed ELCRec framework innovatively combines clustering and intent learning into a single end-to-end differentiable model, resolving the long-standing issue of separate clustering and optimization phases. This integration enhances efficiency and accuracy, representing a significant advancement in the field.
2.	The paper introduces Intent-Coupled Contrastive Learning (ICL), a groundbreaking method that significantly enhances user embeddings by incorporating intent information. This approach addresses the limitations of traditional contrastive learning methods and can substantially improve recommendation systems' performance.
3.	By providing complete experimental code and detailed descriptions of the experimental procedures, the authors ensure that other researchers can easily replicate and validate the results, contributing significantly to the research community.

**Weaknesses:**

1.	In the online A/B test section, the author mentions that this method can efficiently handle new users, but the paper seems do not provide detailed information on how embeddings or cluster centers are assigned to new users during inference.
2.	The paper employs a combined training approach using next_item loss, ICL loss, and cluster loss. However, it does not clarify the appropriate proportions for these three losses, particularly whether the ratio between next_item loss and ICL loss should be fixed at 0.1.
3.	The paper does not explain why the proposed framework results in increased latency on the sports dataset, as observed in Table 2, raising concerns about the consistency and generalizability of the method's performance across different datasets.
4.	The paper claims that existing intention-based methods typically rely on the Expectation-Maximization (EM) algorithm for stepwise training, which leads to suboptimal performance. However, many contemporary works based on Vector Quantization (VQ)[1][2] or Residual Quantization (RQ)[3] have already adopted end-to-end training approaches. The paper does not provide a comparative analysis or discussion of these VQ/RQ-based methods, which is a significant oversight given their relevance and effectiveness in the field.


[1] Hou, Yupeng, et al. "Learning vector-quantized item representation for transferable sequential recommenders." Proceedings of the ACM Web Conference 2023. 2023.
[2] Jiang, Gangwei, et al. "xLightFM: Extremely memory-efficient factorization machine." Proceedings of the 44th International ACM SIGIR Conference on Research and Development in Information Retrieval. 2021.
[3] Rajput, Shashank, et al. "Recommender systems with generative retrieval." Advances in Neural Information Processing Systems 36 (2024).

**Questions:**

1.	According to weakness 1, could you provide more information on how embeddings or cluster centers are determined for new users when they are first introduced to the system?
2.	According to weakness 2, it seems that a clear reason for ratio between next_item losses and ICL losses is required.
3.	According to weakness 3, could you elaborate on the comparative advantages of your method over these VQ/RQ-based methods?
4.	As an addition to table2, could you compare the time and space cost between ELCRec and regular non-clustering recommendation methods?

**Limitations:**

\

---

> ### Author Rebuttal · Authors · 2024-08-07
>
> ## **Response to Reviewer py4J [1/3]**
>
> Thanks for your valuable and constructive reviews. We appreciate your insights and suggestions, as they will undoubtedly contribute to improving the quality of our paper. In response to your concerns, we provide answers to the questions as follows in order.
>
> ### **Cluster Centers for New Users**
> Thanks for your question and suggestion. We first briefly introduce the principles of the baseline and our proposed method in the industrial scenario for a better understanding of the cluster embedding assignment. To conduct personalized recommendations for different user groups, e.g., new users and users with high activities, we adopt the MMOE model and control the gates in the MMOE with the users’ activities to select the approximate experts for the users with different activities. To improve the performance of this baseline model, before inputting the activity embeddings into the gate, we aim to conduct intent learning on users and separate the users into different groups based on their activity features. Then, input the group embeddings to the gates of MMOE, therefore better helping the model to recognize the different user groups, match the experts, and provide a more precise recommendation.
>
> Then, we discuss the user group assignment problem at two different stages of the recommendation. For the recommendation produced by the model, i.e., at the rank stage, it just needs to separate the different user groups and provide personalized recommendations for new users and users with high activities, and it doesn’t need to know which groups are exactly the new user group or the high-activity user group. This way can already provide personalized recommendations for different user groups and solve the cold-start problem in recommendation. Moreover, at the pre-rank stage, we may design some recommendation strategies for different user groups. Therefore, we need to know the clustering assignment of the different user groups. Note that, after training and clustering, we can obtain the clustering assignment of all samples (users). And then we need to label the different user groups based on the user activities or other manual tags of the users by some simple strategies, such as voting and ensemble. After labeling different user groups, we can provide different recommendation strategies, such as boosting or un-boosting for different user groups.
>
> In summary, at the rank stage, there is no need for the model inference to provide the exactly labels for each user groups. Besides, at the re-rank stage, if we want to design some strategies for different user groups, we can adopt the vote or ensemble methods to label the user group embeddings based on their activities or other manual tags of the users. And the case studies in Section 7.9.1 and the performance improvement in Table 4 of Section 5.2 demonstrate that our method can separate the new users and other users with high activities and provide precise personalized recommendation for them, respectively. We add these details in Section 7.13 and highlight them in red in the revised paper: https://anonymous.4open.science/r/NeurIPS-2354-ELCRec-revised-DFCC/NeurIPS24-2354-ELCRec-revised.pdf.
>
> ### **Proportions for Loss Functions**
> Thanks for your question. The balance is set to 1 in equation (7). We can add one balance hyperparameter to control the balance between sequence contrastive learning loss and intent contrastive learning loss to achieve better performance. However, in equation (8), we find there are many balances that need to be controlled, such as the balance of intent-assist contrastive learning loss and the balance of intent learning loss, easily leading to the high cost of hyperparameter tuning. To lower the load of tune hyperparameters, we fix the balance between sequence contrastive learning loss and intent contrastive learning loss as 1 and the balance between next item prediction loss and intent-assisted contrastive learning loss as 0.1. The reason for setting the ratio between next item prediction loss and ICL loss to 0.1 is that we regard the next item prediction task as the core task of the recommendation since it can directly influence the performance of the recommendation, while the ICL loss is regarded as the auxiliary loss function, which guides the network to conduct some pre-text tasks to further improve the quality of the embeddings. And this setting has already been able to achieve promising performance. For other complex scenarios, we can set more balance hyperparameters for better performance in the future. We have revised our paper and provided a detailed discussion about the balance problem in the method part. The revised part is highlighted in red, and for the revised paper, please refer to https://anonymous.4open.science/r/NeurIPS-2354-ELCRec-revised-DFCC/NeurIPS24-2354-ELCRec-revised.pdf.
>
> **to be continue...**

---

> ### Author Response · Authors · 2024-08-07
>
> ## **Response to Reviewer py4J [2/3]**
>
> ### **VQ\RQ-based Methods**
> Thanks for your suggestion. We have carefully read and analyzed these papers. And we briefly introduce them and compare them with our ELCRec to highlight the merits of our proposed method. VQ-Rec [1] is proposed to solve the issues, including over-emphasizing effect of text features and exaggerating the negative impact of domain gap by learning the vector-quantized item representation. The schema of VQ-Rec is summarized as text->code->representation. However, VQ-Rec mainly focusses on the item representation and the number of items is always largely smaller than the number of users in the large-scale recommenders. In addition, in the original paper of VQ-Rec, it mentions “the used technique for training OPQ, i.e., k-means, tends to generate clusters with a relatively uniform distribution on ...”. It seems that VQ-Rec adopts the conventional k-means clustering for the code, therefore may leading to the out-of-memory and long training time problems. Besides, similarly, [2] propose an extremely memory-efficient factorization machine named xLightFM, where each category embedding is composited with latent vectors selected from the codebooks. xLightLM is a factorization-machine-based recommendation method, which is different from the sequential recommendation methods and hard to process the sequence data. Additionally, in the original paper of xLightLM, the authors mentioned “..., which first decomposes the embedding space into the Cartesian product of subspaces and conducts the k-means clustering in each subspace for obtaining center vectors”. It also simply adopts the k-means clustering algorithm on the embedding to obtain the codebooks. Thus, it also meets the out-of-memory and long training time problems on the large-scale data. Moreover, a generative retrieval approach named TIGER [3] is proposed by creating semantically meaningful tuple of codewords to serve as a Semantic ID for each item. Although the residual quantization is verified effective, method seems still based on the offline clustering since the authors mentioned “we use k-means clustering-based initialization for the codebook”. In addition, it also mainly focuses on the item embeddings and aims to provide the semantical information for the items. Different from them, our method mainly focuses on the user embeddings, which are more numerous compared with the items. Also, our proposed method utilizes the end-to-end learnable clustering to unify the intent learning and behavior learning int an unified framework. It not only improves the recommendation performance, but also improve the scalability of the intent learning method. The evidence can be found in the experiment part of the paper. Moreover, these three related papers seem not focus on the intent learning of users. Nevertheless, we are glad to accept your suggestion and make discussions on these interesting methods. We add them in the related work part (Sectoin 7.11.3) in the revised paper: https://anonymous.4open.science/r/NeurIPS-2354-ELCRec-revised-DFCC/NeurIPS24-2354-ELCRec-revised.pdf.
>
>
>     [1] Hou, Yupeng, et al. "Learning vector-quantized item representation for transferable sequential recommenders." Proceedings of the ACM Web Conference 2023. 2023.
>
>     [2] Jiang, Gangwei, et al. "xLightFM: Extremely memory-efficient factorization machine." Proceedings of the 44th International ACM SIGIR Conference on Research and Development in Information Retrieval. 2021.
>
>     [3] Rajput, Shashank, et al. "Recommender systems with generative retrieval." Advances in Neural Information Processing Systems 36 (2024).
>
> **to be continue...**

---

> ### Author Response · Authors · 2024-08-07
>
> ## **Response to Reviewer py4J [3/3]**
>
> ### **Latency on Sports Dataset**
> Thanks for your careful review and question. We observe that in most cases, our proposed method can save time and memory costs, e.g., saving 7.18% time and 9.48% memory on average. For the time cost of our method on the Sports dataset, we regard it as a corner case. By careful analyses, we provide the explanation as follows. We suspect the raised time costs are caused by the wrong direction of the optimization. Setting the cluster embeddings as the learnable neural parameters and optimizing them during training may be a harder task for the model compared to conducting the offline clustering algorithm on the learned embeddings directly. We analyze the performance and loss curve of our method on the Sports dataset, and find that the decline of loss slowdowns and the performance seems drops a little at the almost end of the training. We think this wrong optimization leads to the comparable time cost of our method compared with the baseline. But for other datasets, their optimization processes are great, therefore saving time and memory costs essentially. In the future, we can avoid this wrong optimization direction through some strategies, such as early-stopping and penalty terms. We add this explanation of the corner case in the Section 4.4 of the revised paper: https://anonymous.4open.science/r/NeurIPS-2354-ELCRec-revised-DFCC/NeurIPS24-2354-ELCRec-revised.pdf.
>
>
> ### **Time & Space Costs**
> Thanks for your suggestion. Following your suggestion, we add the conventional self-supervised-learning-based sequential recommendation method S3-Rec in the cost comparison experiments, since ICLRec is based on S3-Rec and comparing other regular methods is not very informative. Due to the limitation time of the rebuttal phase, we only add one method in this experiment, and during the discussion phase, we are willing to compare more methods if you require. The experimental results are demonstrated as follows. We find that the conventional self-supervised-learning-based recommendation method S3-Rec costs more time and memory compared with the ICLRec and ELCRec since 1) it contains two training phases, including the pre-training and the fine-tuning. 2) It incorporates four complex self-supervised learning tasks, including associated attribute prediction, masked item prediction, segment prediction, and masked item prediction.
> We add these experimental results and discussion in Section 7.9 and highlight them in red in the revised paper: https://anonymous.4open.science/r/NeurIPS-2354-ELCRec-revised-DFCC/NeurIPS24-2354-ELCRec-revised.pdf.
>
> | Cost   | Dataset | Sports | Beauty | Toys | Yelp | Average  |
> |:------:|:-------:|:------:|:------:|:----:|:----:|:--------:|
> | Time   | S3-Rec  | 8319   | 4414   | 4452 | 5925 | 5778     |
> |        | ICLRec  | 5282   | 3770   | 4374 | 4412 | 4460     |
> |        | ELCRec  | 5360   | 2922   | 4124 | 4151 | 4139     |
> | Memory | S3-Rec  | 2512   | 2294   | 2975 | 3982 | 2941     |
> |        | ICLRec  | 1944   | 1798   | 2887 | 3671 | 2575     |
> |        | ELCRec  | 1781   | 1574   | 2555 | 3383 | 2328     |

---

> > ### Author Response · Authors · 2024-08-12
> >
> > To: Reviewer py4J
> >
> > Dear Reviewer py4J,
> >
> > Hi Reviewer py4J! We highly appreciate your valuable and insightful reviews. We hope the above response has addressed your concerns. If you have any other suggestions or questions, feel free to discuss them. We are very willing to discuss them with you in this period. If your concerns have been addressed, would you please consider raising the score? It is very important for us and this research. Thanks again for your professional comments and valuable time!
> >
> > We sincerely appreciate your constructive reviews and questions. We provide detailed responses regarding Cluster Centers for New Users, Proportions for Loss Functions, VQ\RQ-based Methods, Latency on Sports Dataset, and Time & Space Costs as above. We hope our responses can effectively address your concerns. If they don't, let's have further discussion now.
> >
> > Besides, if you have any additional suggestion or questions, please do not hesitate to bring them up. We are more than willing to engage in further discussion to improve the quality of this research.
> >
> > If you feel that your concerns have been satisfactorily resolved, we kindly ask you to consider revising your score. Your rating is crucial for us and our research. Thank you once again for your professional comments and the time you have invested!
> >
> > Best wishes,
> >
> > Authors of Submission 2354

---

> ### Author Response · Authors · 2024-08-08
>
> Thank you very much for your precious time and valuable comments. We hope our responses have addressed your concerns. Please let us know if you have any further questions. We are happy to discuss them further. Thank you.
>
> Best regards,
>
> Authors

---

> ### Author Response · Authors · 2024-08-10
> **Follow Up for Reviewer py4J**
>
> Dear Reviewer py4J,
>
> We highly appreciate your valuable and insightful reviews. We hope the above response has addressed your concerns. If you have any other suggestions or questions, feel free to discuss them. We are very willing to discuss them with you in this period. If your concerns have been addressed, would you please consider raising the score? It is very important for us and this research. Thanks again for your professional comments and valuable time!
>
> Best wishes,
>
> Authors

---

> ### Author Response · Authors · 2024-08-14
>
> Dear Reviewer py4J,
>
> Thanks for your efforts in this conference and submission. We understand that time is valuable. As the discussion deadline is approaching, we haven't received feedback from you yet. If we still don't receive anything from you, we may assume our responses solve your concerns well. If you have any other questions, feel free to discuss them now.
>
>
> Best regards,
>
> Authors of Submission 2354

---

### Official Review · Reviewer_bYUb · 2024-07-11

**Soundness:** 4
**Presentation:** 3
**Contribution:** 4
**Rating:** 7
**Confidence:** 5

**Summary:**

This paper aims to improve the optimization paradigm of the existing intent learning methods for recommendation. A novel intent learning method named ELCRec is proposed by unifying behavior representation learning into the end-to-end learnable clustering framework. Experiments, theoretical analyses, and application shown the superiority of ELCRec.

**Strengths:**

- The research topic is practical and meaningful. Intent learning plays an important role in user understanding and recommendation.

- The motivation of improving the alternating optimization is clear and the propose method ingeniously solved this problem by the end-to-end learnable clustering framework.

- The paper is very comprehensive, including experiments, applications, and theoretical analyses. The improvement is significant.

- The code is available, which guarantee the reproducibility.

**Weaknesses:**

- The author should provide more intuitions and insights of designing ELCRec before introducing this method, including but not limited to the challenge discussion, naive proposal, further improvement, and deep-in ideas.

- The authors should provide the precise data and show them in the Figure 1 for the better understanding of the effectiveness of the proposed modules.

- The related work part is too short. The authors should make a more comprehensive survey and discuss on Recommendation in 2024. More details about the related papers are required.
1. The authors assert the efficiency of the proposed method. However, details regarding the devices used and the improvements observed in the A/B testing have not been provided.

2. What’s the update rate of user group embedding? After the clustering, will the results be stored to the database?

3. How to determine the cluster number in the practical scenario? The cluster number seems fixed in this method but it is not reasonable for the large-scale users in practical app.

**Questions:**

1. The authors assert the efficiency of the proposed method. However, details regarding the devices used and the improvements observed in the A/B testing have not been provided.

2. What’s the update rate of user group embedding? After the clustering, will the results be stored to the database?

3. How to determine the cluster number in the practical scenario? The cluster number seems fixed in this method but it is not reasonable for the large-scale users in practical app.

**Limitations:**

Yes

---

> ### Author Rebuttal · Authors · 2024-08-07
>
> ## **Response to Reviewer bYUb [1/2]**
>
> Thanks for your valuable and constructive reviews. We appreciate your insights and suggestions, as they will undoubtedly contribute to improving the quality of our paper. In response to your concerns, we provide answers to the questions as follows in order.
>
> ### **Intuitions and Insights**
> Thanks for your constructive suggestion. Following your suggestion, we add more details of design insights before introducing our proposed ELCRec method. Concretely, we first analyze the challenge of scaling the intent learning methods to large-scale industrial data. The existing intent learning methods always adopt the expectation and maximization framework, where E-step and M-step are conducted alternately and mutually promote each other. However, we find the EM framework is hard to scale to large-scale data since it faces two challenges. First, the clustering algorithm is performed on the full data, easily leading to the out-of-memory problem. Second, the EM paradigm limits performance since it separates the behavior learning process and the intent learning process. To solve these two problems, we aim to propose a new intent learning method for the recommendation task. For the first challenge, our initial idea is to design an online clustering method to update the clustering centers at each step. Specifically, we propose an end-to-end learnable clustering module (ELCM) to solve this problem by setting the clustering center as the learnable neural parameters and the pull-and-push cluster loss functions. In addition, for the second challenge, we aim to integrate the intent learning process into the behavior learning process and optimize them together. Benefitting from setting the cluster centers as the learnable neural parameters, we can utilize them to assist the behavior contrastive learning. Namely, we propose intent-assisted contrastive learning, which not only supports the learning process of online clustering but also unifies behavior learning and intent learning. Therefore, with the above two designs, we can solve the challenges of scaling the intent learning method to large-scale data. We have revised our paper and add these intuitions and insights before introducing the proposed method. The revised part is highlighted in red and for the revised paper, please refer to https://anonymous.4open.science/r/NeurIPS-2354-ELCRec-revised-DFCC/NeurIPS24-2354-ELCRec-revised.pdf.
>
>
> ### **Precise Data**
>
> Thanks for your suggestion. The precise data is provided as follows. We have revised our paper and provided the precise data of the ablation studies in Appendix 7.5. The revised part is highlighted in red and for the revised paper, please refer to https://anonymous.4open.science/r/NeurIPS-2354-ELCRec-revised-DFCC/NeurIPS24-2354-ELCRec-revised.pdf.
>
>
>    |        |    B      | B+ICL  | B+ELCM | ELCRec  |
>    |:------:|:------:|:------:|:------:|:------:|
>    | Sports | 0.1343 | 0.1379 | 0.1396 | 0.1405  |
>    | Beauty | 0.2390  | 0.2398 | 0.2432 | 0.2473  |
>    | Toys   | 0.2664 | 0.2675 | 0.2718 | 0.2686  |
>    | Yelp   | 0.1258 | 0.1262 | 0.1285 | 0.1305  |
>
> ### **Related Work**
> Thanks. In the main text, we merely provide the brief introduction of the related papers due to the page limitation. And we provide the comprehensive and detailed introduction of the related work in the Appendix 7.10. It contains three parts, including sequential recommendation, intent learning for recommendation, and clustering algorithm. And many papers [1-5] published in 2024 have already been surveyed and discussed.
>
>     [1] Dong X, Song X, Liu T, et al. Prompt-based Multi-interest Learning Method for Sequential Recommendation[J]. arXiv preprint arXiv:2401.04312, 2024.
>
>     [2] Li Z, Xie Y, Zhang W E, et al. Disentangle interest trends and diversity for sequential recommendation[J]. Information Processing & Management, 2024, 61(3): 103619.
>
>     [3] Bai Y, Zhou Y, Dou Z, et al. Intent-oriented Dynamic Interest Modeling for Personalized Web Search[J]. ACM Transactions on Information Systems, 2024, 42(4): 1-30.
>
>     [4] Qin X, Yuan H, Zhao P, et al. Intent Contrastive Learning with Cross Subsequences for Sequential Recommendation[C]//Proceedings of the 17th ACM International Conference on Web Search and Data Mining. 2024: 548-556.
>
>     [5] Ma H, Xie R, Meng L, et al. Plug-in diffusion model for sequential recommendation[C]//Proceedings of the AAAI Conference on Artificial Intelligence. 2024, 38(8): 8886-8894.
>
>
> **to be continue...**

---

> ### Author Response · Authors · 2024-08-07
>
> ## **Response to Reviewer bYUb [2/2]**
>
> ### **Devices & Efficiency**
> Thanks for your question. For the devices, we use the company’s self-develop devices and can not provide the details due to commercial reasons. For the A/B testing, we provide the details and the results in Section 5. Concretely, the performance improvement can be found in Table 3 and Table 4. In addition, for efficiency, it’s hard to conduct experiments on real-time large-scale data since the existing intent learning methods will lead to out-of-memory and long-running time problems. But our propose method can solve these problems and scale to the large-scale industrial recommendation system. Besides, to test the efficiency improvement, we conduct detailed experiments on the open benchmarks. The details can be found in Table 2.
>
>
> ### **Group Embeddings**
> Thanks for your questions. The questions regarding the update and the store of group embedding are essential to the deployment of the system. The user group embeddings dynamically change during the training stage. On the open benchmarks, we control the update rate by setting the learning rate of the intent embeddings. Concretely, the learning rate is set to 1e-3, as mentioned in the implementation details. Besides, on the real-time data, in order to better control the update rate, we utilize the momentum update strategy by considering both the current status and historical status of the group embeddings. The details are illustrated in Section 7.13. For the utilization of group embeddings, there are many ways. For the conventional user recommendation or the group recommendation, we utilize the historical group embeddings and conduct continue training for the recommendation model. For other downstream tasks in other domains, we can provide the restore group embeddings for them. Therefore, for the recommendation model, the group embeddings are restored in the model parameters and updated daily. Besides, for other indirect downstream tasks, the group embeddings will be stored in the database. We add these details in Section 5.2 of the revised paper: https://anonymous.4open.science/r/NeurIPS-2354-ELCRec-revised-DFCC/NeurIPS24-2354-ELCRec-revised.pdf.
>
>
>
> ### **Determine Cluster Number**
> Thanks. Determining cluster number is a common challenge in most of clustering methods. In this paper, we set the cluster number as a hyperparameter and conduct hyperparameter experiments in Appendix 7.5. In the future, we can develop a new method to determine the cluster number based on sample density detection, reinforcement learning, etc. This aspect is discussed in Appendix 7.12.

---

> ### Author Response · Authors · 2024-08-08
>
> Thank you very much for your precious time and valuable comments. We hope our responses have addressed your concerns. Please let us know if you have any further questions. We are happy to discuss them further. Thank you.
>
> Best regards,
>
> Authors

---

> ### Author Response · Authors · 2024-08-10
> **Follow Up for Reviewer bYUb**
>
> Dear Reviewer bYUb,
>
> We highly appreciate your valuable and insightful reviews. We hope the above response has addressed your concerns. If you have any other suggestions or questions, feel free to discuss them. We are very willing to discuss them with you in this period. If your concerns have been addressed, would you please consider supporting this paper during the discussion period. Thanks again for your professional comments and valuable time!
>
> Best wishes,
>
> Authors

---

> > ### Comment · Reviewer_bYUb · 2024-08-13
> > **Reply to rebuttal**
> >
> > Thanks for the rebuttals. My concerns have been addressed and the quality of the paper is improved in the revised version. I decide to keep my score.

---

> > > ### Author Response · Authors · 2024-08-13
> > >
> > > Dear Reviewer bYUb,
> > >
> > > Thank you for your professional reviews and valuable suggestions. Your feedback has significantly improved the quality of our paper. We are pleased that our responses have effectively addressed your concerns, and that you are willing to give an acceptance score. Should you have any further questions, we are more than willing to discuss them with you.
> > >
> > > Warm regards,
> > >
> > > Authors of Submission 2354

---

### Official Review · Reviewer_WQpX · 2024-07-12

**Soundness:** 4
**Presentation:** 3
**Contribution:** 4
**Rating:** 6
**Confidence:** 5

**Summary:**

In this paper, the authros study the complex optimization  issue in the filed of recommendation.  It encodes users' behavior sequences and successfully unifies behavior representation learning into  a learnable clustering framework. Further, it uses cluster centers as self-supervision signals to highlight mutual promotion. Experimental results dmonstrate the effectivenss of the proposed method.

**Strengths:**

1. The organization is easy to follow, and related work is fairly comprehensive.

2. Motivation is strong, recommendation is an intersting and practical topic, and this study will facilitate this  field.

3. The unification of behavior representation learning and clustering optimization is novel, and it is conducive to enhancing the optimization paradigm.

**Weaknesses:**

1. In section 5.2, the number of large-scale datasets is  relatively small. The authros are encouraged to add more large-scale datasets so as to further highlight its wide applicability.

2. It would be better to add the summary about the devised loss Eq.(8).

3. Some suboptimal experimental results should be discussed in more detailed, such as Fig. 1 (c).

**Questions:**

1. In Eq.(7), is there no balance hyperparameter for two losses?

2.  When the clustering is not learnable, how is the performance of the proposed ELCRec?  Ablation study is needed.

3.  When the number of clusters is unkonwn, how is the performance of ELCRec?

**Limitations:**

Yes

---

> ### Author Rebuttal · Authors · 2024-08-07
>
> ## **Response to Reviewer WQpX [1/2]**
>
> Thanks for your valuable and constructive reviews. We appreciate your insights and suggestions, as they will undoubtedly contribute to improving the quality of our paper. In response to your concerns, we provide answers to the questions as follows in order.
>
> ### **Large-scale Dataset**
> Thanks for your question. In this paper, we aim to solve the practical problems in the large-scale industrial recommendation system. Therefore, we first design our method and conduct quick experiments on the toy open benchmarks. Then, we conduct extensive experiments on the real-time large-scale data in the application (with about 130 million page views / 50 million user views per day). We admit the scalability of the open benchmarks is limited, but we think it is reasonable for quick trials, and our final aim is to deploy the method in real-world applications.
>
>
> ### **Devised Loss**
> Thanks. Following your suggestion, we detail and summarize the devised loss in equation (8). We train our proposed ELCRec method with multiple tasks, including the next-item prediction task, intent-assisted contrastive learning, and intent learning (learnable clustering) task. Accordingly, Equation (8), which denotes the overall loss function of ELCRec, contains three parts: next-item prediction loss $\mathcal{L}\_{\text{next-item}}$, the intent-assisted contrastive learning loss $\mathcal{L}\_{\text{icl}}$, and the intent learning loss $\mathcal{L}\_{\text{cluster}}$. Concretely, the next-item prediction loss is a commonly used loss function for the sequential recommendation. It aims to predict the next item in the interaction sequence based on the previous sequence. In addition, the intent learning loss aims to optimize the cluster center embeddings by pulling the samples to the corresponding cluster centers and pushing away different cluster centers. Moreover, the intent-assisted contrastive learning loss aims to conduct self-supervised learning to unify the behavior representation learning and intent representation learning. Overall, equation (8) trains the network through three tasks by a linear combination of three loss functions. We have revised our paper and provided the summary of the devised loss in the end of the method part. The revised part is highlighted in red and for the revised paper, please refer to https://anonymous.4open.science/r/NeurIPS-2354-ELCRec-revised-DFCC/NeurIPS24-2354-ELCRec-revised.pdf.
>
> ### **Sub-optimal Results**
> Thanks for your careful review of the experimental results and constructive suggestions. We did have one inconsistent finding on the toy dataset compared with other datasets. Concretely, ELCRec (B+ELCM+ICL) cannot beat B+ELCM, indicating that ICL may be ineffective on the B+ELCM variant on this dataset. However, we also find that B+ICL can beat B, indicating that ICL works for the baseline model. This phenomenon is interesting. We have the following explanations as follows. The ICL is conducted on both the behavior representations and the intent representations. Therefore, it can be influenced by both these two optimization processes. Namely, both the quality of behavior embeddings and the quality of the intent embeddings are crucial for the quality of ICL. Thus, it may not be very robust in all cases. For B+ICL, adding ICL to the baseline can improve the behavior-learning process. However, we find that B+ELCM has already achieved a very promising performance compared with other variants, indicating the quality of intent representations is excellent. Then we add ICL to B+ELCM, the ICL may downgrade the quality of intent representations. To solve this issue, we will conduct more careful training and optimize the training procedure to achieve better performance. We have revised our paper and provided detailed discussion in the Section 4.2. The revised part is highlighted in red and for the revised paper, please refer to https://anonymous.4open.science/r/NeurIPS-2354-ELCRec-revised-DFCC/NeurIPS24-2354-ELCRec-revised.pdf.
>
> **to be continue...**

---

> ### Author Response · Authors · 2024-08-07
>
> ## **Response to Reviewer WQpX [2/2]**
>
> ### **Balance Hyper-parameter**
> Thanks for your question. The balance is set to 1 in equation (7). We can add one balance hyperparameter to control the balance between sequence contrastive learning loss and intent contrastive learning loss to achieve better performance. However, in equation (8), we find there are many balances that need to be controlled, such as the balance of intent-assist contrastive learning loss and the balance of intent learning loss, easily leading to the high cost of hyperparameter tuning. To lower the load of tune hyperparameters, we fix the balance between sequence contrastive learning loss and intent contrastive learning loss as 1 and the balance between next item prediction loss and intent-assisted contrastive learning loss as 0.1. This setting has already been able to achieve promising performance. For other complex scenarios, we can set more balance hyperparameters for better performance in the future. We have revised our paper and provided detailed discussion about the balance problem in the method part. The revised part is highlighted in red and for the revised paper, please refer to https://anonymous.4open.science/r/NeurIPS-2354-ELCRec-revised-DFCC/NeurIPS24-2354-ELCRec-revised.pdf.
>
> ### **Ablation Study**
> Thanks for your question. If the clustering is not learnable, the method will be degraded to the baseline method. Concretely, in our opinion, clustering will not be learnable if we remove the end-to-end learnable clustering method (ELCM) from the variant methods. Correct us here if you have a different understanding of non-learnable clustering, and we can have further discussions. If you agree with this setting, then the corresponding ablation studies are shown in Figure 1, where B, B+ICL, B+ELCM, ELCM denotes the baseline, the baseline with intent-assisted contrastive learning, the baseline with end-to-end learnable clustering method, and the baseline with both. From these experimental results, we can find that 1) B+ELCM can achieve better performance than B. 2) ELCM (B+ICL+ELCM) can beat the B+ICL. Therefore, the effectiveness of the learnable clustering is verified.
>
>
> ### **Unknown Cluster Number**
> Thanks. Determining the cluster number is a common challenge in most clustering methods. In this paper, we set the cluster number as a hyperparameter and conduct hyperparameter experiments in Appendix 7.5. In the future, we can develop a new method to determine the cluster number based on sample density detection, reinforcement learning, etc. This aspect is discussed in Appendix 7.12.

---

> ### Author Response · Authors · 2024-08-08
>
> Thank you very much for your precious time and valuable comments. We hope our responses have addressed your concerns. Please let us know if you have any further questions. We are happy to discuss them further. Thank you.
>
> Best regards,
>
> Authors

---

> ### Author Response · Authors · 2024-08-10
> **Follow Up for Reviewer WQpX**
>
> Dear Reviewer WQpX,
>
> We highly appreciate your valuable and insightful reviews. We hope the above response has addressed your concerns. If you have any other suggestions or questions, feel free to discuss them. We are very willing to discuss them with you in this period. If your concerns have been addressed, would you please consider raising the score? It is very important for us and this research. Thanks again for your professional comments and valuable time!
>
> Best wishes,
>
> Authors

---

### Author Rebuttal · Authors · 2024-08-07

We extend our sincere gratitude to the SAC, AC, and PCs for their dedicated efforts and constructive feedback. Your comments have been invaluable in enhancing the quality of our manuscript. We have meticulously addressed each of your questions and hope our responses satisfactorily address your concerns. According to your suggestions, we have revised our paper and highlighted the revised part in red. Please refer to the following anonymous link: https://anonymous.4open.science/r/NeurIPS-2354-ELCRec-revised-DFCC/NeurIPS24-2354-ELCRec-revised.pdf.

---

> ### Author Response · Authors · 2024-08-13
>
> Dear (Senior) Area Chairs,
>
> With the discussion deadline fast approaching, we kindly request your assistance in reminding the reviewers to provide their post-rebuttal responses. We have spared no efforts to address their concerns.
>
> Currently, the all of the reviewers including Reviewer WQpX, Reviewer bYUb, and Reviewer py4J have not yet responded to the authors. If they have any further questions or concerns, we remain dedicated to addressing them with the utmost eagerness.
>
> Thanks for your help!
>
> Best regards,
> Authors of Submission 2354

---

### Decision · Program_Chairs · 2024-09-25

**Decision:**

Accept (poster)

**Comment:**

Reviews are mostly aligned in favor of accepting. Reviewers praise the motivation and that the paper is comprehensive with strong experiments and available code. One reviewer is more negative, and all reviewers complain about certain details of the experiments, or parts of the paper that feel unintuitive or lack explanation. A slight mitigating factor on this one is that the two most positive reviews are also the most cursory, compared to the more negative review which seems more detailed.